# Environmental signals rather than layered ontogeny imprint the function of type 2 conventional dendritic cells in young and adult mice

Nikos E. Papaioannou[1,2], Natallia Salei[1,2], Stephan Rambichler[1,2], Kaushikk Ravi[1,2], Jelena Popovic [1,2], Vanessa Küntzel[1,2], Christian H. K. Lehmann [3,4], Remi Fiancette[5], Johanna Salvermoser[1,2], Dominika W. Gajdasik[5], Ramona Mettler[1,2], Denise Messerer[6], Joana Carrelha[7], Caspar Ohnmacht[8], Dirk Haller [9,10], Ralf Stumm [11], Tobias Straub [12], Sten Eirik W. Jacobsen[7,13], Christian Schulz [6,14], David R. Withers [5], Gunnar Schotta [15,16], Diana Dudziak[3,4,17,18] & Barbara U. Schraml [1,2✉]

Conventional dendritic cells (cDC) are key activators of naive T cells, and can be targeted in adults to induce adaptive immunity, but in early life are considered under-developed or functionally immature. Here we show that, in early life, when the immune system develops, cDC2 exhibit a dual hematopoietic origin and, like other myeloid and lymphoid cells, develop in waves. Developmentally distinct cDC2 in early life, despite being distinguishable by fate mapping, are transcriptionally and functionally similar. cDC2 in early and adult life, however, are exposed to distinct cytokine environments that shape their transcriptional profile and alter their ability to sense pathogens, secrete cytokines and polarize T cells. We further show that cDC2 in early life, despite being distinct from cDC2 in adult life, are functionally competent and can induce T cell responses. Our results thus highlight the potential of harnessing cDC2 for boosting immunity in early life.

[1] Faculty of Medicine, Biomedical Center, Institute for Cardiovascular Physiology and Pathophysiology, LMU Munich, 82152 Planegg-Martinsried, Germany. [2] Walter-Brendel-Centre of Experimental Medicine, University Hospital Munich, LMU Munich, 82152 Planegg-Martinsried, Germany. [3] Laboratory of Dendritic Cell Biology, Department of Dermatology, University Hospital Erlangen, Friedrich-Alexander University of Erlangen-Nürnberg, Erlangen, Germany. [4] Medical Immunology Campus Erlangen, University Hospital of Erlangen, Friedrich-Alexander-University (FAU) of Erlangen-Nürnberg, Erlangen, Germany. [5] Institute of Immunology and Immunotherapy, College of Medical and Dental Sciences, University of Birmingham, Birmingham, UK. [6] Medizinische Klinik und Poliklinik I, University Hospital Munich, LMU Munich, 81377 Munich, Germany. [7] Haematopoietic Stem Cell Biology Laboratory, MRC Molecular Haematology Unit, MRC Weatherall Institute of Molecular Medicine, Radcliffe Department of Medicine, University of Oxford, Oxford, UK. [8] Center of Allergy and Environment, Helmholtz Center and Technical University of Munich, 80802 Munich, Germany. [9] ZIEL–Institute for Food and Health, Technische Universität München, Freising, Germany. [10] Chair of Nutrition and Immunology, Technische Universität München, Freising, Germany. [11] Institute of Pharmacology and Toxicology, Jena University Hospital, Jena, Germany. [12] Core Facility Bioinformatics, Biomedical Center, LMU Munich, 82152 Planegg-Martinsried, Germany. [13] Department of Medicine Huddinge, Center for Hematology and Regenerative Medicine and Department of Cell and Molecular Biology, Karolinska Institutet, Stockholm, Sweden. [14] DZHK (German Center for Cardiovascular Research), Partner Site Munich Heart Alliance, 80802 Munich, Germany. [15] Faculty of Medicine, Center for Integrated Protein Science Munich, LMU Munich, 82152 Planegg-Martinsried, Germany. [16] Faculty of Medicine, Division of Molecular Biology, Biomedical Center, LMU Munich, 82152 Planegg-Martinsried, Germany. [17] Deutsches Zentrum Immuntherapie (DZI), Erlangen, Germany. [18] Comprehensive Cancer Center Erlangen-European Metropolitan Area of Nuremberg (CCC ER-EMN), Erlangen, Germany. ✉email: barbara.schraml@bmc.med.lmu.de

Vaccination is a reliable means of inducing protective immunity but it can have limited efficacy in infants[1]. The newborn immune system exhibits quantitative and qualitative differences compared to adult hosts that ultimately result in dampened immune responses in early life, which are necessary to tolerate the sudden encounter with commensal microbes and environmental antigens[2,3]. The unique features of the early life immune system also leave infants at increased risk of infection[3]. Antigen presentation and priming of T cells are pre-requisites for adaptive immune responses and the establishment of protective immunity. Improving T-cell priming by tailoring the design of vaccines towards the age-specific immune parameters of neonates and infants has therefore been suggested as a possibility to boost vaccination efficacy and immunity in early life[3,4].

Classical or conventional dendritic cells (cDCs) are potent activators of naive T cells that have successfully been targeted to induce adaptive immune responses in adults[5–8]. In early life, however, these cells are considered under-developed or functionally immature. In murine and human neonates, the cDC compartment is smaller than that of adults[9–11] and cDCs from both human and murine neonates express lower basal levels of major histocompatibility complex II (MHCII) and costimulatory molecules rendering them less potent at stimulating T cells[12–15]. The early life murine cDC compartment further exhibits an intrinsic bias to generate T-helper type 2 (Th2) immune responses due to delayed production of IL-12p70[16,17]. Such Th2 bias is also evident in cord blood cDCs, which exhibit reduced IL-12p70 production and upregulation of costimulatory molecules following pathogenic stimulation compared to cDCs from peripheral blood of adults[13,14,18]. However, expansion of cDCs in early life by administration of the DC growth factor fms tyrosine kinase 3 ligand (FLT3L) can improve innate and adaptive immune defense in mice[19,20], indicating a potential of early life cDCs to initiate functional immune responses.

The cDC compartment is composed of developmentally distinct subsets with unique functions in immunity[5,6,21]. cDC1 are potent cross-presenters and activators of CD8$^+$ T cells and promote Th1 differentiation through the production of interleukin-12 (IL-12) and interferon-γ (IFN-γ)[22–24]. In contrast, cDC2 are considered more potent activators of CD4$^+$ T cells and inducers of Th2, Th17, and T-follicular helper cell differentiation[25–29]. The neonatal cDC compartment in mice is dominated by cDC1, whereas cDC2 constitute the main cDC subtype in most adult organs[9,10,30,31]. Considering the predominant role of cDC1 in producing IL-12 and IFN-γ in adults[23,24], it is unlikely that subset distribution can account for the observed Th2 bias of the cDC compartment in early life and in most human tissues the relative frequency of cDC1 and cDC2 remains stable throughout life[15]. These data suggest that an additional layer of age-dependent regulation influences cDC function. This is highlighted by the fact that neonatal cDC1 are refractory to IFN-α signaling, produce less IL-12 but more IL-10 than their adult counterparts and have a reduced capacity to activate CD4$^+$ T cells and select epitopes for presentation to CD8$^+$ T cells during viral infection[9,16,17,32,33].

Few studies have investigated cDC2 in early life. In neonatal mouse spleen, CD8$^-$ cDCs show lower production of IFN-γ and reduced activation of alloreactive T cells than their adult counterparts[9]. Although CD8 expression does not reliably distinguish cDC1 and cDC2 in neonatal spleen[17,34], these data suggest a reduced ability of neonatal cDC2 to activate T cells. Similarly, cDC2 in neonatal lung express lower costimulatory molecules than adult cDC2[32,35–37], although they can promote Th2-mediated allergy and induce some CD8$^+$ T-cell proliferation when infected with respiratory syncytial virus[30,32,38]. These data indicate that in early life, cDC2 have the potential to activate T cells in some circumstances, raising the question whether this capacity could be harnessed in a broader sense, for instance to initiate T-cell responses in the spleen, which is a major site for antibody production[39].

The first blood and immune cells arise from extra-embryonic yolk sac progenitors and for some cell types, such as macrophages or mast cells, yolk sac-derived cells can persist in tissues for extended periods of time after birth[40–42]. cDCs first arise during embryogenesis in mice and humans, when they are thought to critically contribute to fetomaternal tolerance[9,10,40,43]. However, the origins of cDCs in early life have not been investigated. In adults, cDCs develop downstream of hematopoietic stem cells (HSCs) from a fraction of myeloid bone marrow progenitors capable of generating plasmacytoid DCs (pDCs) and cDCs that has been termed common DC progenitor (CDP)[44–47]. Within CDPs in mice, expression of the C-type lectin receptor DNGR-1 (encoded by the *Clec9a* gene) distinguishes cDC-restricted progenitors[48]. CDPs further differentiate into pre-cDCs, which continue to express DNGR-1[48], exit the bone marrow and terminally differentiate into cDC subtypes in peripheral organs and in response to environmental cues[49]. The *Clec9a* promoter is also active in cDC1 and to a lower extent on pDCs but it is not active in cDC2 and other lymphoid or myeloid cells[48,50–52]. By crossing mice expressing CRE-recombinase under the *Clec9a* promoter to *Rosa*$^{lox-stop-lox}$-yellow fluorescent protein (YFP) or *Rosa*$^{lox-stop-lox}$-TOMATO mice, we have generated mice that faithfully track cells belonging to the cDC lineage in steady state and under inflammatory conditions[48,53,54]. Using these mice, we have found that cDC2 in adult spleen are predominantly derived from CDP/pre-cDC[48,53,54], which is consistent with other studies[55,56].

Here we set out to investigate the differences between cDC2 in early and adult life and clarify the reasons underlying age-dependent functional regulation of cDC2. We reveal that cDC2, analogous to other lymphoid and myeloid cell types, develop in waves. cDC2 arising from distinct hematopoietic sources in early life are phenotypically and transcriptionally similar, and functionally capable to induce T-cell responses. cDC2 in early and adult life, however, have a distinct ability to sense pathogens, produce cytokines and induce T-cell differentiation that is transcriptionally imprinted by distinct cytokine signaling in early and adult life. Exploiting the unique aspects of cDC2 in early life may thus provide an efficient means to boost vaccination efficacy and protective immunity.

## Results

**Early-life cDC2 exhibit ontogenetic diversity.** To investigate phenotype and origin of cDCs in early life, we first profiled these cells in *Clec9a*$^{cre/+}$*Rosa*$^{TOM}$ mice. In this context, we defined the neonatal period as the first 10 days after birth[57]. In the steady-state mouse spleen, cDCs, identified as CD11c$^+$MHCII$^+$ cells[5], could be found as early as embryonic day 16 (E16, Fig. 1a, b and Supplementary Fig. 1). The frequency of CD11c$^+$MHCII$^+$ cells increased steadily with age and reached adult levels at around 4 weeks of age (Fig. 1a, b and Supplementary Fig. 1), confirming results of previous studies[9,10]. CD11c$^+$MCHII$^+$ cells could further be divided into CD24$^+$ and CD11b$^+$ cells[5]. CD24$^+$ cDCs include XCR-1$^+$ BATF3-dependent cDC1, as well a subset of non-canonical BATF3-independent CD8$^+$ cDCs that expresses CD172a and CX3CR1 but lacks XCR-1 (Fig. 1a)[58–60]. As expected, CD24$^+$ cDCs dominated the splenic cDC compartment in early life but became less frequent with age, whereas the proportion of CD11b$^+$ cDC2 increased with age[9] (Fig. 1c). Notably, XCR-1$^+$ cDC1 were the more frequent cell type within CD24$^+$ cells at most ages examined, although in 2-day-old and adult mice non-canonical CD8$^+$ cDC and cDC1 were present at equal

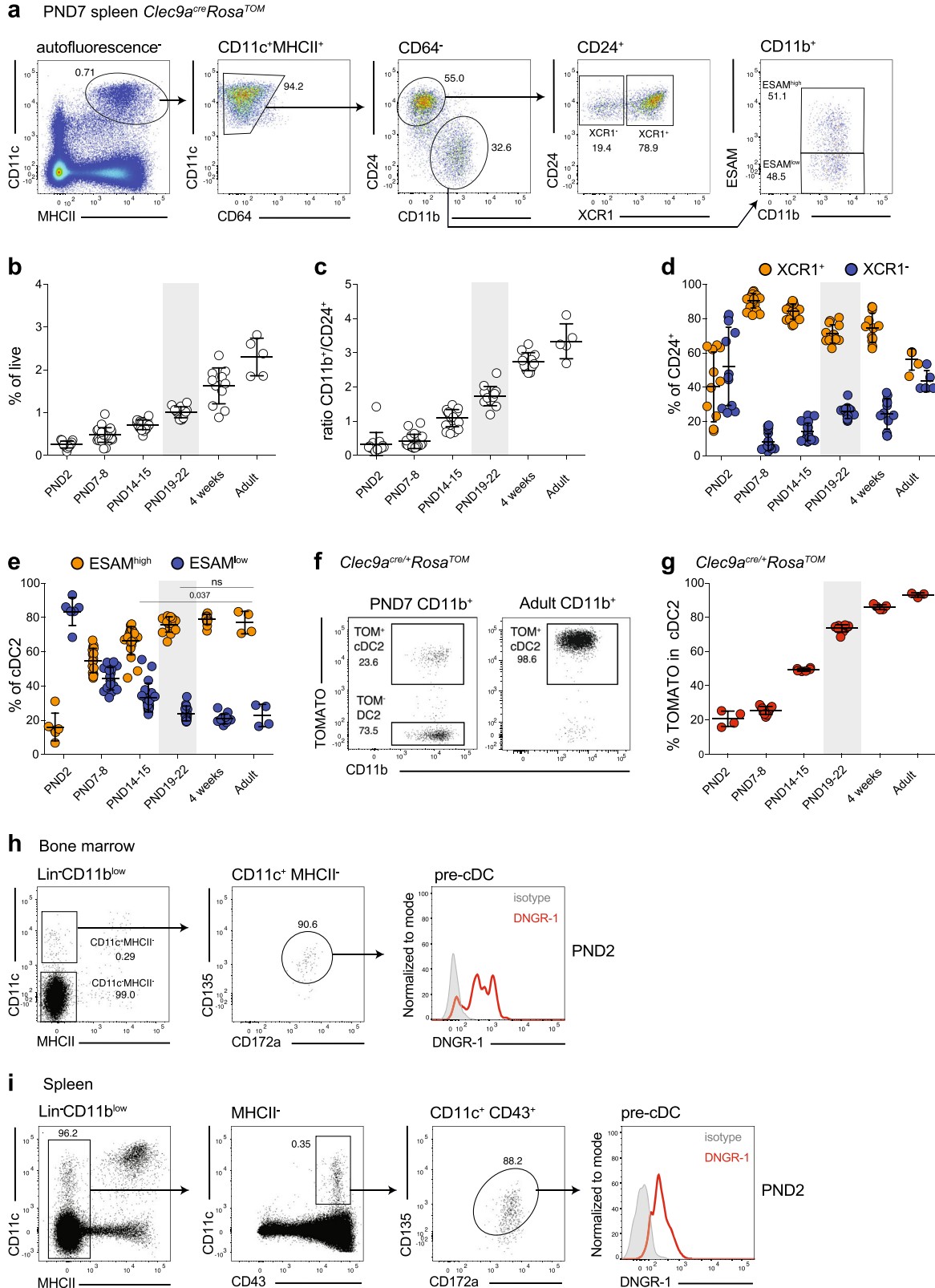

proportions (Fig. 1d). In the spleen, CD11b$^+$ cDC2 can be divided into NOTCH2- and RUNX3-dependent ESAM$^{high}$ cDC2 and ESAM$^{low}$ cDC2, which develop independent of these transcription factors[61–63] (Fig. 1a and Supplementary Fig. 1A, B). Notably, ESAM$^{high}$ cells were less frequent within the cDC2 compartment of neonatal compared to adult mice; however, the proportion of ESAM$^{high}$ cells within the cDC2 compartment reached adult levels between 2 and 3 weeks of age (Fig. 1e and Supplementary Fig. 1A, B), which coincides with the organization of the splenic architecture.

We next assessed *Clec9a*-expression history with age by profiling TOMATO expression in splenic leukocytes from *Clec9a$^{cre/+}$Rosa$^{TOM}$* mice. cDC2 lack DNGR-1 expression and therefore TOMATO labeling in this population is a true indicator

**Fig. 1 Developmental heterogeneity of early life cDC2. a–e** Spleens from *Clec9a*[cre/+]*Rosa*[TOM] and *Clec9a*[cre/cre]*Rosa*[TOM] mice of the indicated ages were analyzed by flow cytometry. **a** Single live autofluorescence-negative cells were gated and cDCs identified as CD11c[+]MHCII[+]CD64[−] cells. cDCs were further divided into CD24[+] and CD11b[+] cells, and analyzed for XCR-1 and ESAM expression, respectively. **b** Frequency of CD11c[+]MHCII[+] cDCs in total splenocytes (n = 13, PND2; n = 25, PND7–8; n = 20, PND14–15; n = 14, PND19–22; n = 12, 4 weeks; n = 5, adult). **c** Ratio of CD11b[+] to CD24[+] cells within CD11c[+]MHCII[+] cDCs (n = 13, PND2; n = 21, PND7–8; n = 20, PND14–15; n = 14, PND19–22; n = 12, 4 weeks; n = 5, adult). **d** The percentage of XCR-1[+] and XCR-1[−] cells within CD24[+] cDC1 is shown (n = 12, PND2; n = 21, PND7–8; n = 20, PND14–15; n = 14, PND19–22; n = 12, 4 weeks; n = 5, adult). **e** The percentage of ESAM[high] and ESAM[low] cells within CD11b[+] cDC2 is shown (n = 6, PND2; n = 11, PND7–8; n = 18, PND14–15; n = 14, PND19–22; n = 11, 4 weeks; n = 4, adult). **f** Representative flow cytometry gating for TOMATO expression in splenic cDC2 from 1-week-old and adult *Clec9a*[cre/+]*Rosa*[TOM] mice. **g** The frequency of TOMATO[+] cells within splenic CD11b[+] cDC2 of *Clec9a*[cre/+]*Rosa*[TOM] mice with age (n = 4, PND2; n = 11, PND7–8; n = 6, PND14–15; n = 10, PND19–22; n = 5, 4 weeks; n = 3, adult). Each dot represents one mouse, horizontal bars represent mean, error bars represent SD, gray rectangles indicate weaning. ns: not significant. Statistical analysis was performed using two-tailed *t*-test. **h, i** Bone marrow (**h**) and spleen (**i**) from 2-day-old wild-type mice were analyzed by flow cytometry. Lin[−](CD3, B220, NK1.1, CD4, CD8α, TER119) cells were gated, pre-cDCs were identified as indicated and stained with anti-DNGR-1 (red) or isotype-matched control antibodies (gray). Source data are provided as a Source Data file.

of cell origin[48]. Consistent with our previous observations and their CDP origin[5,48,54–56], cDC2 from adult spleen showed near-complete labeling with TOMATO in *Clec9a*[cre/+]*Rosa*[TOM] mice (93 ± 1.28%). Notably, in neonatal mice TOMATO labeling in cDC2 was strongly reduced (20.6 ± 4.48%), indicating ontogenetic heterogeneity. The frequency of TOMATO[+] cells within cDC2 increased with age until adult labeling was reached at around 4 weeks of age (86 ± 1.4%; Fig. 1f, g).

We have previously shown that increased CRE levels (in homozygous *Clec9a*[cre] mice) leads to markedly increased labeling of DC precursors and differentiated cDCs, because a fraction of precursors 'escapes" labeling in a stochastic fashion[48]. In addition, the choice of fate reporter can influence labeling efficiency due to differences in the distance between the loxP sites[64]. To exclude that reduced labeling of neonatal cDC2 was an artifact of the specific fate reporter used or due to "progenitor escape" in mice heterozygous for *cre*, we analyzed homozygous *Clec9a*[cre/cre]*Rosa*[TOM] mice and *Clec9a*[cre/cre]*Rosa*[YFP] mice. In both models, a relative lack of *Clec9a*[cre]-mediated fate labeling in neonatal cDC2 was observed (Supplementary Fig. 1C, D). Importantly, TOMATO labeling did not correlate with either of the two cDC2 subsets, as ESAM[high] or ESAM[low] cells showed similar labeling frequency at all ages examined (Supplementary Fig. 1E). Comparable to observations in adult mice[48,54], *Clec9a*[cre]-mediated TOMATO labeling in young mice remained restricted to CD11c[+]MHCII[+] cDCs and cells resembling pDCs or pre-cDCs based on low CD11c and MHCII expression (Supplementary Fig. 1F, G).

As most of our work on DC ontogeny stems from studies in adult mice, we next confirmed that DNGR-1-expressing cDC progenitors were present in early life bone marrow and spleen. Cells resembling macrophage dendritic cell progenitors (MDPs) (lineage (lin)[−] CD11c[−]MHCII[−]CD11b[low]CD115[+]CD135[+]CD117[high]), CDPs (lin[−]CD11c[−]MHCII[−]CD11b[low]CD115[+]CD135[+]CD117[low]), and pre-cDCs (lin[−]MHCII[−]CD11b[low]CD11c[+]CD135[+]CD172a[int]) could be found in bone marrow from neonatal mice (Supplementary Fig. 2A, B). Notably, putative MDPs were more frequent, whereas CDP and pre-cDC-like cells were less frequent in neonatal compared to adult bone marrow (Supplementary Fig. 2B). Importantly, pre-cDC-like cells in bone marrow from 2-day- and 1-week-old mice uniformly expressed DNGR-1 (Fig. 1h and Supplementary Fig. 2C). Accordingly, TOMATO was detected in CDPs and pre-cDCs, but not MDPs or common lymphoid progenitors (CLPs) of 1- and 2-week-old *Clec9a*[cre/cre]*Rosa*[TOM] mice, as defined phenotypically (Supplementary Fig. 2E). These data confirm that *Clec9a*-driven CRE was active in DC precursors in early life. Importantly, pre-cDCs from 1- and 2-week-old mice labeled with TOMATO to a similar extent as their counterparts from adult mice, although labeling at 1 week of age had not completely reached adult levels (Supplementary Fig. 2E;

81.15 ± 2.77% PND7, 94.67 ± 1.72% 2 weeks, 93.61 ± 2.93% adult). Pre-cDC-like cells with unimodal DNGR-1 expression were also found in spleen from 2-day- and 1-week-old mice (Fig. 1i and Supplementary Fig. 2D), and labeled with TOMATO at similar levels as their bone marrow counterparts (Supplementary Fig. 2E, F), indicating that CRE-mediated labeling of pre-cDCs completes in bone marrow. Actively cycling progenitor cells can escape labeling because of a time lag between CRE protein synthesis and DNA recombination[45,48,49]. If cDC2 in neonates escaped labeling because of such rapid transition through their precursor stage, we would expect higher labeling in differentiated cDC2 than in their immediate progenitors. However, splenic pre-cDC labeled with TOMATO to a larger extent than differentiated cDC2 from the same mice at one and 2 weeks of age (Supplementary Fig. 2F). This is in contrast to adult mice, where labeling of splenic cDC2 and pre-cDCs was similar (Supplementary Fig. 2f). To further exclude the possibility that cDC2 in early life lack detectable TOMATO, because they have had insufficient time to accumulate enough fluorescent protein, we sorted TOMATO[+] and TOMATO[−] CD11c[+]MHCII[+]CD11b[+] cells from 1-week-old *Clec9a*[cre/cre]*Rosa*[TOM] mice and cultured them in the presence of the DC survival factors FLT3L and Granulocyte-macrophage colony-stimulating factor (GM-CSF). After 24 h, TOMATO[−] cells had remained TOMATO[−], supporting the notion that these cells arise independently of *Clec9a*-expressing progenitors (Supplementary Fig. 2G). Thus, cells resembling bona fide *Clec9a*-expressing cDC progenitors exist in early life and *Clec9a*[cre] mice can be used to trace cDCs in early life.

**Fate mapping reveals a lymphoid contribution to cDC2 in early life.** To address the origin of cDC2 in early life, we first crossed *Clec9a*[cre/cre]*Rosa*[YFP] mice to mice lacking the growth factor FLT3L, which is required for the development of all DCs[65]. In 2-week-old *Clec9a*[cre/cre]*Rosa*[YFP]*Flt3l*[−/−] mice, both cDC1 and cDC2 were strongly reduced (Fig. 2a). Thus, early-life cDC2 require FLT3L for their development, independent of whether they arise from *Clec9a*-expressing progenitors or not. Hematopoiesis begins before birth and, in mice, yolk sac progenitors contribute to the generation of lymphoid and myeloid immune cells[41,42,66,67]. To assess the contribution of yolk sac erythromyeloid progenitors to the cDC2 compartment in early life, we treated pregnant *Csf1r*[Mer-iCre-Mer]*Rosa*[YFP] dams at E8.5 with 4OH-tamoxifen[41,67] and analyzed spleens from offspring mice at E18.5, as well as 2 and 4 weeks after birth. Importantly, neither CD24[+] nor CD11b[+] cDCs labeled with YFP at any ages examined, when compared to liver Kupffer cells and splenic macrophages, which served as positive controls (Fig. 2b and Supplementary Fig. 3A). The transcription factor c-Myb is dispensable for yolk sac hematopoiesis but essential for the

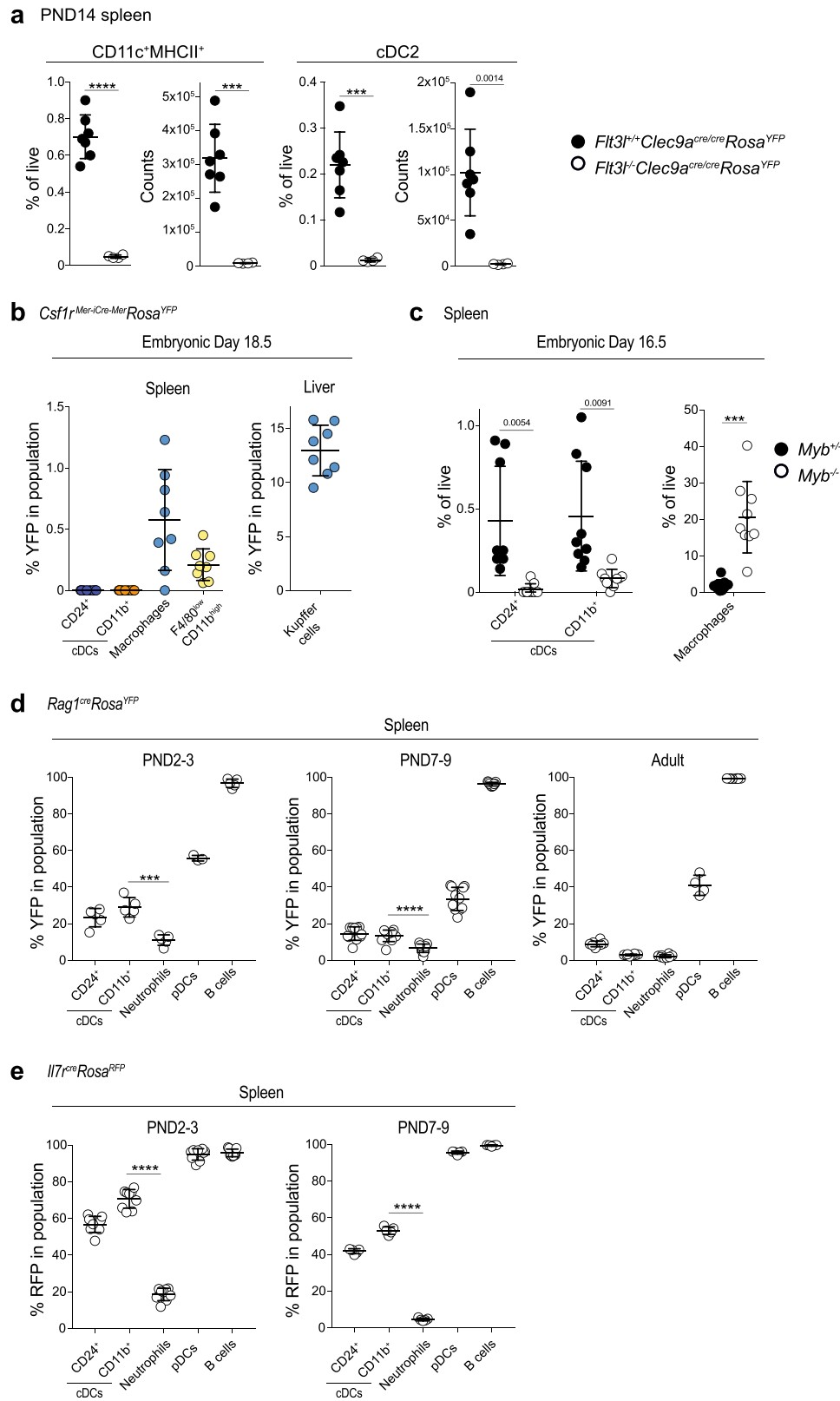

development of definitive HSCs[68]. *Myb*-deficient mice are embryonically lethal but survive until E16.5. At this time CD24[+] and CD11b[+] cDCs, but not splenic macrophages, were reduced in spleens from *Myb*[−/−] compared to littermate control embryos (Fig. 2c). Thus, CD11c[+]MHCII[+] cells arise independent of yolk

sac-derived erythromyeloid progenitors and in late embryogenesis require c-Myb for their development, suggesting that they arise from definitive hematopoiesis.

CLPs can generate cDC-like cells in adoptive transfer[69–73] and we have shown that lymphopoiesis contributes to cDC generation

**Fig. 2 Fate mapping reveals a lymphoid contribution to cDC2 in early life. a** Spleens from 2-week-old $Flt3l^{-/-}Clec9a^{cre/cre}Rosa^{YFP}$ ($n = 7$) and $Flt3l^{+/+}Clec9a^{cre/cre}Rosa^{YFP}$ ($n = 4$) mice were analyzed by flow cytometry. Shown are the frequency and number of CD11c$^+$MHCII$^+$ cells (left) and of cDC2 (middle). **b** $Csf1r^{Mer-iCre-Mer}$ dams were mated with male $Rosa^{YFP}$ mice and injected with 4OH-tamoxifen on E8.5. Spleen and liver from offspring mice were analyzed by flow cytometry on E18.5. The percentage of YFP$^+$ cells in the indicated populations is plotted ($n = 8$). **c** Spleens from $Myb^{-/-}$ and $Myb^{+/-}$ littermate control mice were analyzed on E16.5. The frequency of the indicated populations was calculated and plotted ($n = 9$). **d** The percentage of YFP$^+$ cells within the respective populations in spleen from $Rag1^{cre}Rosa^{YFP}$ mice at the indicated ages ($n = 5$, PND2–3; $n = 11$ PND7–9; $n = 8$ adult; $n = 3$ and 4 for pDCs in PND2–3 and adult mice). **e** The percentage of RFP$^+$ cells within the indicated populations in spleen from $Il7r^{cre}Rosa^{RFP}$ mice at the indicated ages ($n = 9$, PND2–3; $n = 5$, PND7–9). Each dot represents one mouse, horizontal bars represent mean, error bars represent SD. ***$p < 0.001$, ****$p < 0.0001$. Statistical analysis in **a**, **c**–**e** was performed using two-tailed $t$-test. Source data are provided as a Source Data file.

when myeloid cDC progenitors are impaired[74]. When cultured in vitro with FLT3L CLPs from $Clec9a^{cre/+}Rosa^{TOM}$ mice gave rise to CD11c$^+$MHCII$^+$ cDC-like cells that predominantly resembled cDC1. However, few cDC2-like cells were also generated. These remained TOMATO negative, suggesting that CLP can generate TOM$^-$ cDC2 (Supplementary Fig. 3B). Expression of recombination-activating gene-1 is first detected in CLPs and required for the development of B and T cells by promoting somatic rearrangements of antigen receptor genes[75]. Therefore, we profiled YFP labeling of splenic leukocytes in neonatal $Rag1^{cre}Rosa^{YFP}$ mice at 2–3 and 7–9 days after birth, as well as in adulthood. As lineage decisions are stochastic, fate mapping with constitutive Cre is not absolute and should be considered at the population level and in the context of positive and negative control populations. Although B cells exhibited near-complete labeling with YFP at all ages examined (Fig. 2d), labeling in neutrophils ($11 \pm 2.86\%$ PND2–3, $6.63 \pm 2.32\%$ PND7–9, $2.27 \pm 0.82\%$ adults, Fig. 2d) and bone marrow erythroid cells ($5.32 \pm 1.88\%$ PND2–3, $1.87 \pm 0.8\%$ PND7–9) (Supplementary Fig. 3C), which served as negative controls, was low as expected[66]. Notably, in 2- to 3-day- and 7- to 9-day-old $Rag1^{cre}Rosa^{YFP}$ mice cDC2 labeled with YFP at $29 \pm 5.30\%$ and $13.43 \pm 3.16\%$, respectively (Fig. 2d and Supplementary Fig. 3C), whereas in cDC2 from adult spleen $Rag1^{cre}$-mediated YFP labeling did not exceed labeling of negative control populations (Fig. 2d and Supplementary Fig. 3C). Importantly, YFP labeling was comparable between ESAM$^{high}$ ($16.15 \pm 1.01\%$) and ESAM$^{low}$ ($15.21 \pm 1.07\%$) cDC2 in 7- to 9-day-old mice (Supplementary Fig. 3D), again supporting similar early life heterogeneity in both subsets. Of note, CD24$^+$ cDC1 showed greater evidence of $Rag1^{cre}$ expression history compared to cDC2 in neonatal and adult mice (Fig. 2d). IL-7 receptor ($Il7r$)-cre mice have been used to demonstrate that lymphoid progenitors do not contribute to cDCs in adult mice[76]. We therefore used these mice as independent model to map the fate of lymphoid-restricted progenitors. Importantly, cDC2 from 2-day- and 1-week-old $Il7r^{cre}Rosa^{RFP}$ mice labeled strongly with RFP, although labeling was lower in cDC2 from 1 week compared to 2-day-old mice (Fig. 2e).

$Cxcr4^{creER}$ mice label consecutive stages of definitive hematopoiesis in embryonic and adult mice, while not labeling yolk sac progenitors[77]. We therefore pulsed $Cxcr4^{creER}Rosa^{mTmG}$ mice with tamoxifen at E12.5, reasoning that fetal liver HSCs would be labeled, allowing us to assess their contribution to splenic DCs. At E18.5, microglia, which served as negative control, exhibited no green fluorescent protein (GFP) labeling, as expected[77]. In contrast, we observed labeling of monocytes and macrophages in E18.5 spleen, consistent with their descendance from fetal liver HSCs (Supplementary Fig. 3E). Importantly, cDCs labeled with GFP to a similar extent, supporting the notion that fetal liver-resident progenitors contribute to cDC2 in early life (Supplementary Fig. 3E).

Thus, fate mapping indicates a lymphoid contribution to the steady-state cDC2 pool in early life, but not adulthood. Coupled to the observation that cDC2 acquire $Clec9a^{cre}$ expression history with age, these data suggest that cDC2 development is regulated in waves, with fetal liver-resident lymphoid progenitors contributing early but being gradually replaced with age by bona fide cDCs arising from $Clec9a$-expressing progenitors. Henceforth, we will refer to CD11c$^+$MHCII$^+$CD11b$^+$ cells from $Clec9a^{cre}Rosa^{TOM}$ mice that lack TOMATO expression as TOM$^-$ DC2 (Fig. 1f).

**TOM$^+$ cDC2 and TOM$^-$ DC2 in early life are phenotypically similar.** Phenotypic analysis revealed no obvious differences in the expression of the prototypical cDC2 markers CLEC4A4, CD172a, CD26, ESAM, or CD4 on TOM$^+$ cDC2 and TOM$^-$ DC2 from 1-week-old $Clec9a^{cre/cre}Rosa^{TOM}$ mice (Fig. 3a). Expression of MHCII and the costimulatory molecule CD80 were also similar between TOM$^+$ and TOM$^-$ cells (Fig. 3a). Using cytospin analyses we found cells with the typical dendritic morphology of cDCs, as well as cells with round morphology and little cytoplasm within TOM$^+$ cDC2 and TOM$^-$ DC2 (Supplementary Fig. 3F). TOM$^+$ cDC2 and TOM$^-$ DC2 in spleens from 1-week-old mice were located in the developing white pulp and in close proximity to CD3$^+$ T cells (Fig. 3b). Thus, $Clec9a$-negative progenitors generate cDC2-like cells with similar phenotype and tissue localization to bona fide cDC2 arising from $Clec9a$-expressing progenitors.

**TOM$^+$ cDC2 and TOM$^-$ DC2 in early life are transcriptionally identical.** Although environment strongly impacts cell identity, myeloid and lymphoid cell types arising from distinct hematopoietic sources at specified times during development can differ transcriptionally and functionally[42,78–82]. To address if ontogeny transcriptionally imprints cDC2, we compared the gene expression profile of TOM$^+$ cDC2 and TOM$^-$ DC2 from 1-week-old $Clec9a^{cre/cre}Rosa^{TOM}$ mice to that of TOM$^+$ cDC2 from adult mice. We used homozygous $Clec9a^{cre/cre}Rosa^{TOM}$ mice to ensure highest penetrance $Clec9a^{cre}$ and the 1-week time point, because ontogenetic heterogeneity was prominent enough to sort sufficient cells of both populations (Supplementary Fig. 1C), although ESAM$^{high}$ to ESAM$^{low}$ subset distribution at this age was still slightly lower than in adult mice (Fig. 1e). In principal component analysis (PCA) TOM$^+$ cDC2 and TOM$^-$ DC2 from 1-week-old mice segregated away from adult TOM$^+$ cDC2, identifying age as a major contributor to variation (Fig. 4a). Principle component 2 allowed some segregation between TOM$^+$ cDC2 from TOM$^-$ DC2 from young mice, indicating differences in gene expression correlating with differential ontogeny (Fig. 4a). Using unsupervised hierarchical k-means clustering of differentially expressed genes we defined 14 clusters with distinct expression characteristics (Supplementary Fig. 4A). Of these, most clusters identified differences between cells from young and adult mice independent of $Clec9a^{cre}$ expression history (Supplementary Fig. 4A), again highlighting that the biggest differences in gene expression are related to age. Clusters 8 and 14 identified genes

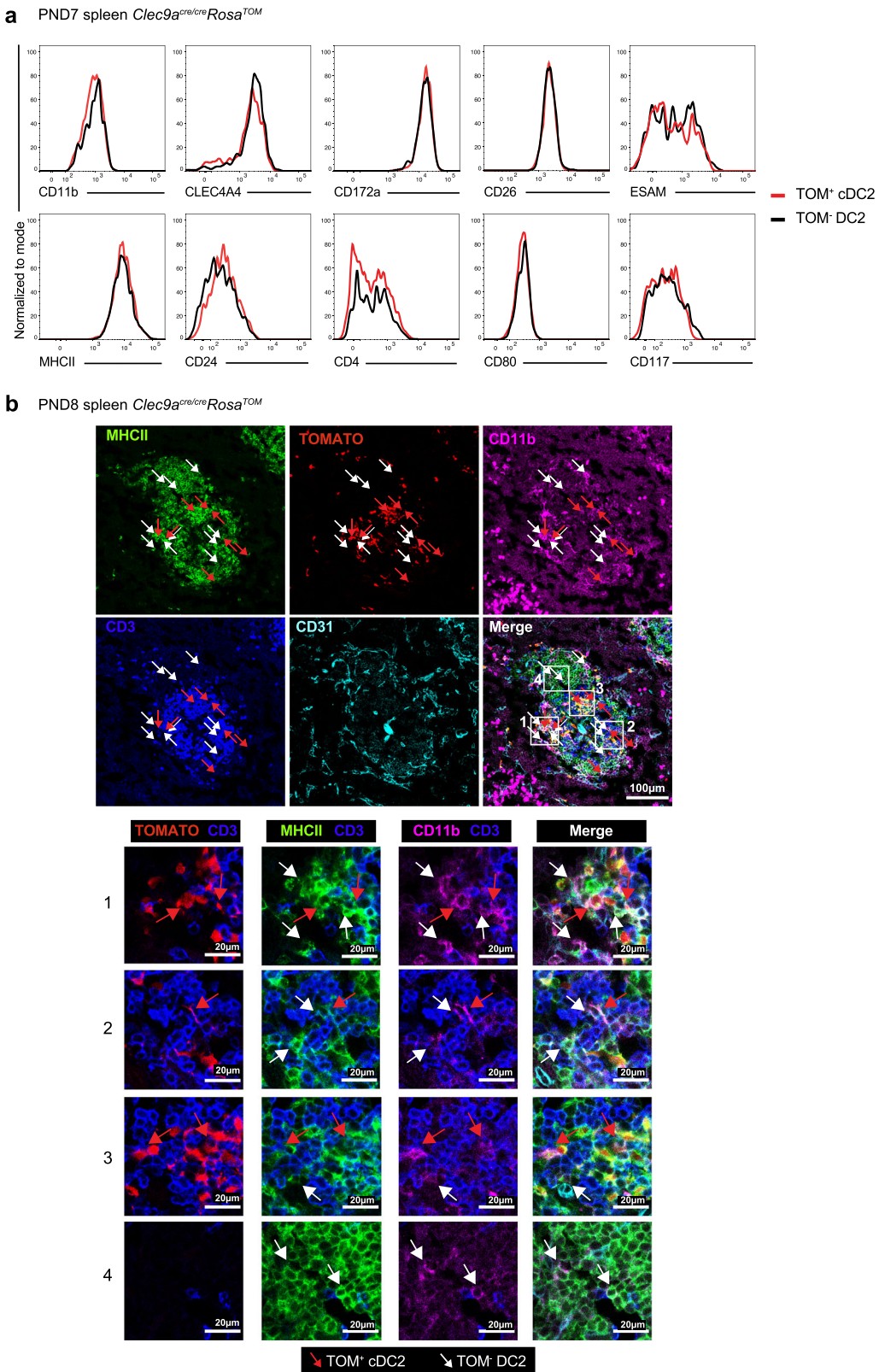

**Fig. 3 TOM⁻ DC2 phenotypically resemble TOM⁺ cDC2. a** TOM⁺ cDC2 and TOM⁻ DC2 from 1-week-old *Clec9a^{cre/cre}Rosa^{TOM}* mice were analyzed for expression of the indicated surface markers. Data are representative of at least two independent experiments with four to seven mice. **b** Spleens from PND8 *Clec9a^{cre/cre}Rosa^{TOM}* mice were analyzed for expression of CD31 (cyan), CD3 (blue), MHCII (green), TOMATO (red), and CD11b (magenta) by microscopy. Numbered inset squares were magnified on the bottom panels TOM⁺ cDC2 (red arrows) or TOM⁻ DC2 (white arrows) were identified. Data are representative of two independent experiments with three biological replicates.

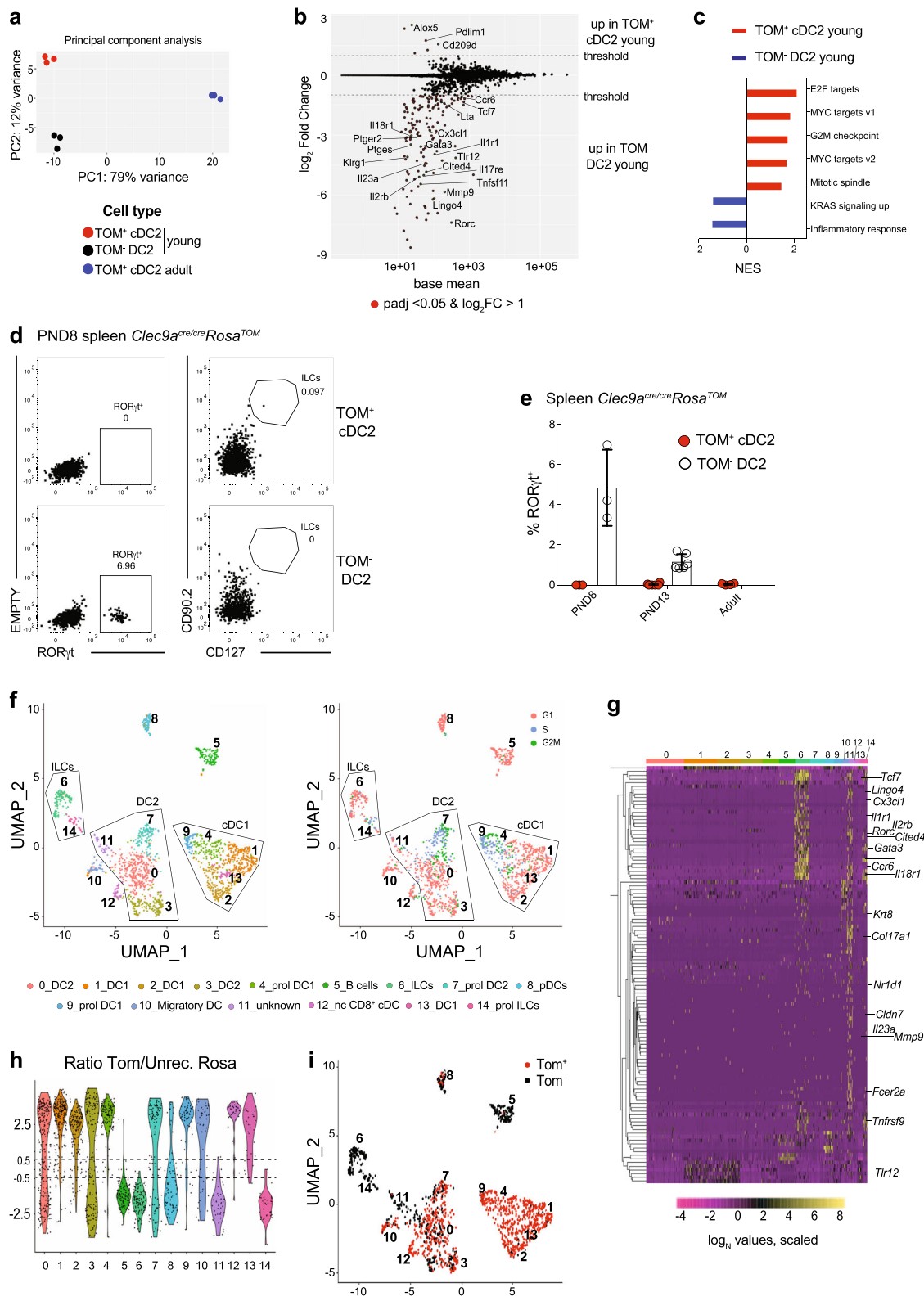

with distinct expression characteristics in TOM⁻ DC2 compared to TOM⁺ cDC2 from young and adult mice (Supplementary Fig. 4A), identifying putative variation caused by ontogeny.

Pairwise comparison of TOM⁺ cDC2 and TOM⁻ DC2 from young mice identified 167 differentially expressed genes ($\log_2$ fold change ($\log_2$FC) > 1, adjusted $p$-value (padj) < 0.05) (Fig. 4b and Supplementary Data 1). In this comparison, TOM⁺ cDC2 showed

an enrichment of cell cycle genes (Fig. 4c), possibly reflecting the fact that these cells expand in number and gradually replace TOM⁻ DC2 with age. Most notable, several genes implicated in the biology of innate lymphoid cells (ILCs), such as *Rorc*, *Tcf7*, *Il18r1*, *Ccr6*, *Lingo4*, and *Lta*, showed higher expression in TOM⁻ DC2 than TOM⁺ cDC2 (Fig. 4b). Using flow cytometry, we detected a fraction of RORγt-expressing cells in TOM⁻ DC2 but

**Fig. 4 TOM⁺ cDC2 and TOM⁻ DC2 in early life are transcriptionally identical. a–c** TOM⁻ DC2 and TOM⁺ cDC2 from 8-day-old and TOM⁺ cDC2 from adult *Clec9a^{cre/cre}Rosa^{TOM}* mice were sorted and analyzed by mRNA sequencing. **a** Principal component analysis (PCA) of the top 500 most variable genes across all samples. Dots represent biological replicates. **b** Pairwise comparison of TOM⁺ cDC2 and TOM⁻ DC2 from 8-day-old mice. **c** Normalized enrichment score (NES) of gene sets enriched in TOM⁺ cDC2 (red) and TOM⁻ DC2 (blue) from 8-day-old mice. **d** TOM⁻ DC2 and TOM⁺ cDC2 from 8-day-old *Clec9a^{cre/cre}Rosa^{TOM}* mice were analyzed by flow cytometry for RORγt, CD90.2, and CD127 expression. Data are representative of three biological replicates. **e** The percentage of RORγt⁺ cells within TOM⁺ cDC2 and TOM⁻ DC2 in *Clec9a^{cre/cre}Rosa^{TOM}* mice at the indicated ages is shown (n = 3, PND8; n = 6, PND13; n = 4, adult). Each dot represents one mouse, horizontal bars represent mean, error bars represent SD. **f, g** CD19⁻MHCII⁺ cells from spleen of 9-day-old *Clec9a^{cre/cre}Rosa^{TOM}* mice were sorted after gating out F4/80^{high} macrophages and subjected to scRNA-Seq. **f** UMAP display of 1788 cells analyzed by unsupervised graph-based clustering with Seurat algorithm to indicate cluster identity (left) and cell cycle status (right) within clusters. **g** Heatmap displaying expression of genes enriched in TOM⁻DC2 compared to TOM⁺ cDC2 in bulk mRNA sequencing (Fig. 4b) among scRNA-Seq clusters. **h, i** The ratio of normalized *Tomato* reads per cell to normalized reads per cell of the predicted transcript of the unrecombined ROSA locus was calculated (**h**) and cells with a ratio > 0.5 were identified as *Tom⁺*, whereas cells with a ratio <−0.5 were identified as *Tom⁻*. **i** *Tom⁺* and *Tom⁻* cells visualized on the UMAP display. Source data are provided as a Source Data file.

not in TOM⁺ cDC2 from 1-week-old *Clec9a^{cre/cre}Rosa^{TOM}* mice (Fig. 4d, E). These cells lacked the canonical ILC markers CD127 and CD90[83–85] and expressed typical cDC2 markers, including CD11c, MHCII, CD172a, CLEC4A4, ESAM and CD26, unlike ILCs (lin⁻CD11b⁻CD90⁺CD127⁺), which are CD11c negative (Supplementary Fig. 4B, C). Notably, the frequency of RORγt⁺ cells within the TOM⁻ DC2 population decreased with age and RORγt⁺ cells were not found in the adult cDC2 pool (Fig. 4e), raising the possibility that these RORγt⁺ cells may constitute a unique population of DC2 that is transiently present in early life and disappears with age.

To address this, we next performed droplet-based single-cell RNA-sequencing (scRNA-seq). We sort-purified splenic MHCII⁺ cells from 9-day-old *Clec9a^{cre/cre}Rosa^{TOM}* mice, after excluding CD19⁺ B cells and F4/80^{high} macrophages (Supplementary Fig. 5A). Using this approach, we expected to identify all main DC subsets, including cDC1, non-canonical CD8⁺ cDCs, TOM⁺ cDC2, and TOM⁻ DC2 and CD11c^{low}MHCII^{low} pDCs, as well as a fraction of RORγt-expressing type 3 ILCs that expresses MHCII but lacks CD11c (Supplementary Figs. 4B and 5a). Unsupervised graph-based clustering identified 15 clusters as visualized by dimensionality reduction using Uniform Manifold Approximation and Projection (UMAP) (Fig. 4f)[86,87]. We identified some proliferating cells (being in G2M and S phase) in all clusters, but excluded clusters 4, 7, 9, and 14 from comparative analyses, as they consisted predominately of proliferating cells (Fig. 4f). The identity of each cluster was determined using the top differentially expressed genes in combination with a priori knowledge about signature genes of cDCs and other immune cells (Supplementary Fig. 5B–E). Cluster 5 identified a contamination with *Cd19*-expressing B cells (Supplementary Fig. 5B), indicating a possible sort impurity or a discord between RNA and protein expression[88]. cDC1 were partitioned across 5 clusters (1, 2, 4, 9, and 13) based on expression of *Xcr1*, *Cd24a*, *Irf8*, and *Tlr3* (Supplementary Fig. 5B, C). We further added the sequence of *cre* as inserted in the *Clec9a* locus[48,89] into the reference transcriptome and found *cre* expression exclusively in clusters 1, 2, 4, 9, and 13, consistent with *Clec9a* expression in early life cDC1[17]. Cluster 8 constituted *Siglech* and *Ccr9* expressing pDCs and clusters 6 and 14 showed high expression of *Rorc*, *Rora*, *Il7r*, *Il18r1*, and *Tox*, identifying them as ILC3s[83,84]. Cluster 12 resembled E2-2-dependent non-canonical CD8⁺ cDCs based on *Cx3cr1*, *Cd24a*, *Sirpa* (CD172a), and *Tcf4* (encoding for E2-2) expression[58,60] (Supplementary Fig. 5B, C). cDC2 were spread across clusters 0, 3, and 7 based on expression of *Sirpa*, *Itgam*, *Irf4*, and published signature genes[90] (Supplementary Fig. 5B, D). Notably, cluster 3 more closely resembled ESAM^{low}, whereas cluster 0 resembled ESAM^{high} cDC2[59] (Supplementary Fig. 5F). In addition, we identified cluster 10 as cells resembling migratory cDCs[90] (Supplementary Fig. 5E). We were

not able to assign a clear identity to cluster 11, although this cluster showed expression of core cDC2 and ESAM^{high} cDC2 signature genes (Supplementary Fig. 5D, F). Notably, *Rorc* was found among the genes characteristic for cluster 11 and this population showed an enrichment of genes with higher expression in TOM⁻ DC2 in the pairwise comparison of TOM⁺ cDC2 and TOM⁻ DC2 in bulk RNA-seq (Fig. 4g and Supplementary Fig. 5B).

We next identified TOM⁺ cDC2 and TOM⁻ DC2 by adding the sequence of *Tomato* and the predicted transcript of the unrecombined *Rosa* locus into the reference transcriptome[91]. Forming the ratio of normalized *Tomato* reads per cell to normalized reads per cell of the predicted transcript of the unrecombined *Rosa* locus (ratio Tom/unrec. Rosa) clearly separated B cells (cluster 5) and cDC1 (clusters 1, 2, 4, 9, and 13), which served as negative and positive controls for *Tomato* expression, respectively (Fig. 4h). Accordingly, we set a threshold and identified *Tom⁻* (ratio Tom/unrec. Rosa < −0.5) and *Tom⁺* cells (ratio Tom/unrec. Rosa > 0.5, Fig. 4h). *Tom⁺* and *Tom⁻* cells distributed evenly within the UMAP of cDC2 clusters 0, 3, and 7 (Fig. 4i), and pairwise comparison of *Tom⁺* and *Tom⁻* cells within clusters 0 and 3 revealed few differences in gene expression, indicating that these cells are transcriptionally similar (Supplementary Data 2). Notably, cells within cluster 11 were predominantly *Tom⁻*, again indicating that this cluster corresponds to RORγt-expressing TOM⁻ DC2 (Fig. 4i).

In adults, cDC2 can be divided into T-bet expressing cDC2A and T-bet-negative cDC2B, putatively controlled by RORγt and C/EBPα[92]. This division appeared to hold up in early life as cluster 0 showed an enrichment of T-bet⁺ cDC2A signature genes (Supplementary Fig. 5G), whereas cluster 3 more closely resembled T-bet⁻ cDC2B (Supplementary Fig. 5G). Interestingly, cells in cluster 11 did not correspond to T-bet⁻ cDC2B but more closely resembled T-bet⁺ cDC2A (Supplementary Fig. 5G), consistent with the high expression of ESAM on RORγt⁺ DC2 (Supplementary Fig. 4C). To gain insights into a putative developmental hierarchy between *Rorc*-expressing cluster 11, cDC2 (clusters 0, 3, and 7), and ILC3 (clusters 6 and 14), we applied Palantir algorithm[93]. Setting the starting point within cluster 11 identified two terminal states, one in ILC cluster 14 and one in cDC2 cluster 0 (Supplementary Fig. 5H). However, differentiation potential diminished quickly upon exit from cluster 11 and the branch probability that cells from cluster 11 reach the terminal states was low (Supplementary Fig. 5H). Accordingly, fate mapping in *Rorc^{cre}Rosa^{RFP}* mice[94] identified only few RFP⁺ cDC2 in 2-week-old mice that included a fraction that also stained positive for RORγt (Supplementary Fig. 5I). As RFP⁺ cells were only a minor fraction of the total splenic DC2 compartment in young mice, RORγt⁺ cells are unlikely to act as progenitors for TOM⁻ DC2 in early life.

Thus, in concordance with fluorescence-activated cell sorting (FACS) analyses, scRNA-seq revealed that cDC2 in 1-week-old mice contain the two main cDC2 subtypes also found in adults, and that, despite their distinct origin, TOM$^+$ cDC2 and TOM$^-$ DC2 distribute evenly across cDC2 clusters, indicating they constitute transcriptionally identical cells. Differences in gene expression between TOM$^+$ cDC2 and TOM$^-$ DC2 from 1-week-old mice in bulk RNA-seq correlated to a unique *Rorc*-expressing cluster of cDC2 that transcriptionally closely resembles but is distinct from ESAM$^{high}$ cDC2 and present exclusively in early life.

**Age defines the strongest differences in gene expression**. Having established ontogeny as a minor contributor to transcriptional variation, we next focused on differences in gene expression caused by age. Pairwise comparison of TOM$^+$ cDC2 from 1-week-old and adult mice identified 1490 differentially expressed genes (log2FC > 1, padj < 0.05) (Fig. 5a and Supplementary Data 3). TOM$^+$ cDC2 from 1-week-old mice showed an enrichment of genes involved in cell cycle (Fig. 5b), possibly reflecting increased homeostatic proliferation within the expanding DC pool. TOM$^+$ cDC2 from adults were enriched for genes implicated in the inflammatory response, which could indicate an increased level of activation or functional maturation of adult cDC2[15]. Most notable was the identification of genes involved in signaling downstream of IFN-γ, tumor necrosis factor-α (TNF-α), IL-2, and IFN-α (Fig. 5b) enriched in cDC2 from adult mice. As expression of receptors for these cytokines was comparable in 1-week-old and adult mice (Supplementary Data 3), these data suggested a distinct cytokine environment acting on cDC2 in spleen from neonatal and adult mice. We confirmed higher expression of PD-L1 (encoded by the *Cd274* gene) and CD38 (*Cd38*), which are regulated by type I and II IFN signaling[95,96], on TOM$^+$ cDC2 from adult compared to 2-week-old mice (Fig. 5c). Importantly, expression of these markers was reduced on cDC2 from adult *Ifnar$^{-/-}$* compared to wild-type control mice (Fig. 5d), supporting the notion that distinct cytokine environments act on splenic cDC2 in early and adult life and indicating that age-dependent differences in gene expression are at least in part caused by IFN-α.

Among genes differentially expressed with age, we identified several pattern recognition receptors (PRRs) with higher expression in TOM$^+$ cDC2 from adults, including *Clec7a* (encoding for Dectin-1), *Tlr7*, and *Tlr5* (Supplementary Fig. 6A). In contrast, expression of *Clec4n* (encoding for Dectin-2), *Tlr4*, and *Tlr2* was higher in early life (Supplementary Fig. 6A), whereas expression of other PRRs, such as *Tlr6*, *Tlr9*, and *Nod1* was comparable in cDC2 from 1-week-old and adult mice (Supplementary Fig. 6A). In line with previous observations in the lung[32,35–37], cDC2 from adult mice showed higher expression of several costimulatory molecules, including *Cd80*, *Cd40*, *Cd274* (encoding for PD-L1) and *Tnfsf4* (encoding for OX40L) (Fig. 5a). Lower expression of costimulatory molecules could indicate a reduced ability of early-life cDC2 to activate T cells but costimulatory signals also balance effector T-cell responses. *Tnfsf4* and *Cd274* for instance can suppress IL-17A production in T cells[97,98]. Interestingly, adult cDC2 further showed higher expression of bone morphogenic protein 2 (*Bmp2*), which also suppresses IL-17A production from T cells[99] (Supplementary Fig. 6B). On the contrary, TOM$^+$ cDC2 from 1-week-old mice showed higher levels of factors that promote Th17 and Treg differentiation, such as *Lgals3*, *Il6ra*, *Lgals1*, and *Sema4a*[100–106] (Supplementary Fig. 6B). In line with age being the foremost contributor to transcriptional variation, expression of the aforementioned genes was similar between TOM$^+$ cDC2 and TOM$^-$ DC2 from 1-week-old mice. Taken together, these data suggest that distinct cytokine environments in early and adult life act on cDC2 to shape their transcriptional profile and ability to induce immune responses.

**Early-life cDC2 induce distinct T-cell responses in vitro compared to adult-life cDC2**. The above data suggested that cDC2 in early and adult life may differ in their ability to induce effector T-cell differentiation. To address this possibility in vitro, we sort-purified TOM$^+$ cDC2 and TOM$^-$ DC2 from young and TOM$^+$ cDC2 from adult mice, and pulsed them with Ovalbumin (OVA) peptide 323–339 (OVA$_{323–339}$). cDC2 populations were sorted from two to 2.5-week-old mice, because at this age ESAM$^{high}$ to ESAM$^{low}$ subset distribution had reached adult levels and ESAM$^{high}$ cells dominated the cDC2 compartment (Fig. 1e). Although DNGR-1 has no known function in DC development and is not expressed in cDC2, we used mice heterozygous for *cre* in functional assays, because *Clec9a$^{cre/cre}$* mice lack functional DNGR-1[48]. Peptide pulsed DC2 populations were subsequently cultured with naive OT-II transgenic T cells from adult mice in the absence or presence of T-cell polarizing cytokines. Adult T cells were chosen as responders, because neonatal T cells exhibit an intrinsic Th2 bias[2]. Notably, TOM$^+$ cDC2 and TOM$^-$ DC2 from young mice and TOM$^+$ cDC2 from adult mice stimulated similar proliferation of naive T cells in all conditions tested (Fig. 5e, f and Supplementary Fig. 6C, D), indicating that adult cDC2 do not exhibit an increased level of activation in terms of their ability to stimulate T cells. Compared to adult cDC2, TOM$^+$ cDC2, and TOM$^-$ DC2 from young mice also induced similar effector differentiation of OT-II cells under non-polarizing (Th0) and Th1 conditions, as assessed by IFN-γ production (Fig. 5g and Supplementary Fig. 6D). We did not observe IL-4 production from T cells stimulated under Th0 conditions by intracellular staining but Th2 cytokine levels in culture supernatants from T cells stimulated with cDC2 from young and adult mice were similar (Supplementary Fig. 6F). Thus, early life splenic cDC2 do not exhibit an intrinsic Th2 bias, which is in contrast to neonatal cDC2 from lung[30]. In line with higher expression of positive regulators of Th17 differentiation, TOM$^+$ cDC2 and TOM$^-$ DC2 from young mice induced twofold higher IL-17A production from T cells under Th17 conditions than cDC2 from adult mice (Fig. 5f, g). Similarly, under Treg conditions more OT-II T cells were positive for Foxp3 when stimulated with DC populations from young mice (Fig. 5f, g). Thus, despite exhibiting increased expression of proliferation related genes (Fig. 5b), cDC2 from young mice induce similar proliferation of T cells as their adult counterparts, whereas inducing higher Th17 and Treg differentiation, indicating qualitative, rather than quantitative differences in the ability to polarize T cells.

As ESAM$^{high}$ and ESAM$^{low}$ cDC2 have different transcriptional programs and may stimulate different types of T-cell responses[62], we next addressed whether both subsets have different functions with age. We therefore sorted ESAM$^{high}$ and ESAM$^{low}$ TOM$^+$ cDC2 from 2-week-old and adult mice and assessed their ability to stimulate Th17 and Treg differentiation as above. ESAM$^{high}$ and ESAM$^{low}$ cDC2 from young mice induced higher Th17 differentiation than their counterparts from adult mice (Fig. 5h). Similarly, ESAM$^{high}$ cDC2 from young mice induced higher Treg differentiation than their adult counterparts, whereas ESAM$^{low}$ cDC2 from young and adult mice induced similar Treg differentiation (Fig. 5h). Thus, functional differences between early and adult life exist for ESAM$^{high}$ and ESAM$^{low}$ cDC2, although there appears to be some level of subset specific functional regulation. As ESAM$^{high}$TOM$^-$ DC2 contain a unique fraction RORγt-expressing cells of unknown function (1.15 ± 0.38%, Fig. 4e), we next crossed *Rorc-eGFP* mice

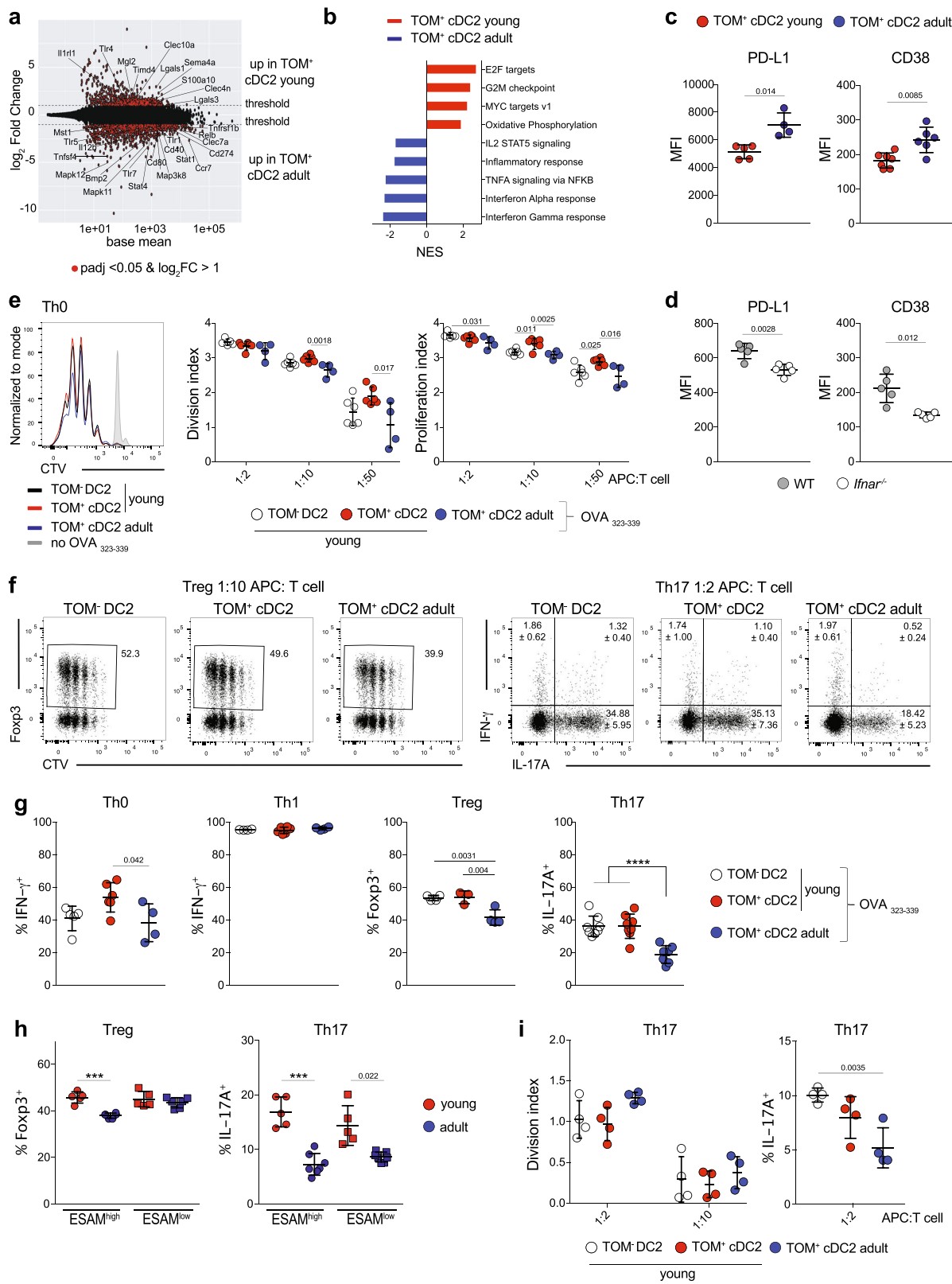

to *Clec9a^{cre/+}Rosa^{TOM}* mice. We then sorted ESAM^{high}TOM^+ cDC2 and ESAM^{high}TOM^−DC2 from 2-week-old *Rorc^{eGFP}Clec9a^{cre/+}Rosa^{TOM}* mice, whereas excluding GFP^+ cells and compared their ability to promote Th17 and Treg differentiation. Importantly, despite excluding RORγt^+ cells ESAM^{high}TOM^+ cDC2 and ESAM^{high}TOM^−DC2 had similar ability to induce Th17 and Treg differentiation (Supplementary Fig. 6G),

confirming that the presence RORγt^+ cells does not influence the ability of TOM^− cDC2 to induce T-cell differentiation.

To determine whether early-life cDC2 are also capable of antigen uptake and processing, we first cultured TOM^+ cDC2 and TOM^− DC2 from 2-week-old and TOM^+ cDC2 from adult *Clec9a^{cre/+}Rosa^{Tom}* mice with fluorescently labeled latex beads in vitro for 2 h in the presence or absence of cytochalasin D. All

**Fig. 5 Age causes strongest differences in gene expression and cell function. a, b** Bulk mRNA sequencing performed as in Fig. 4A–C. **a** Pairwise comparison of TOM$^+$ cDC2 from 8-day-old and adult mice indicating genes with a log$_2$FC > 1 and padj < 0.05 in red. **b** NES of gene sets enriched in TOM$^+$ cDC2 from 8-day-old (red) or in cDC2 from adult mice (blue). **c** TOM$^+$ cDC2 from 2-week-old and adult Clec9a$^{cre/+}$Rosa$^{TOM}$ mice were analyzed for expression of PD-L1 ($n = 5$, 2-week-old; $n = 4$, adult) and CD38 ($n = 7$, 2-week-old; $n = 6$, adult) (MFI = mean fluorescence intensity). **d** Splenic cDC2 from adult wild-type and Ifnar$^{-/-}$ mice were profiled for expression of PD-L1 and CD38 ($n = 5$). Expression levels are depicted as MFI. Each dot represents one mouse, horizontal bars represent mean, error bars represent SD. **e–g** TOM$^-$ DC2 and TOM$^+$ cDC2 from 2-week-old and TOM$^+$ cDC2 from adult Clec9a$^{cre/+}$Rosa$^{TOM}$ mice were sorted, pulsed with OVA$_{323-339}$, and co-cultured with CTV-labeled OT-II cells under Th0 or polarizing conditions for 3.5 days. **e** Left: CTV dilution of OT-II cells co-cultured with the indicated populations or CD11c-enriched splenocytes without OVA$_{323-339}$ (gray). Right: division and proliferation indexes of OT-II cells after co-culture with the indicated DC2 populations ($n = 5/6$, 2-week-old TOM$^+$ and TOM$^-$ cDC2, $n = 4$, adult TOM$^+$ cDC2). **f** T cells were analyzed for expression of Foxp3 and IL-17A. For Th17 conditions (right) numbers in each quadrant represent the mean ± SD ($n \geq 7$). **g** The percentage of cytokine or Foxp3-positive cells within proliferated OT-II cells is shown. APC : T-cell ratio 1 : 2 for Th0, Th1, Th17 and 1 : 10 for Treg ($n = 6$, Th0/Th1; $n = 3$, Treg; $n = 8$, Th17; for 2-week-old TOM$^+$ cDC2; $n = 5$, Th0; $n = 4$, Treg/Th1; $n = 8$, Th17 for TOM$^-$ DC2; $n = 4$, Th0/Th1/Treg ; $n = 7$, Th17 for adult TOM$^+$ cDC2). **h** TOM$^+$ cDC2 from 2-week-old ($n = 5$) and adult mice ($n = 6$, Treg; $n = 7$, Th17) were sorted as ESAM$^{high}$ and ESAM$^{low}$ cells, pulsed with OVA$_{323-339}$ and cultured with CTV-labeled OT-II cells under Treg and Th17 polarizing conditions as above. The percentage of cytokine or Foxp3-positive cells within proliferated OT-II cells is shown. **i** TOM$^-$ DC2 and TOM$^+$ cDC2 from 2-week-old and TOM$^+$ cDC2 from adult Clec9a$^{cre/+}$Rosa$^{TOM}$ mice were co-cultured with OT-II cells in the presence of OVA under Th17 conditions for 4 days. The division index (left) and percentage of IL-17A producing OT-II cells (right, APC : T-cell ratio 1 : 2) are shown ($n = 4$). Each dot represents one biological replicate from four independent experiments, horizontal bars represent mean, error bars represent SD. ***$p < 0.001$, ****$p < 0.0001$. Statistical analysis was performed using two-tailed paired $t$-test (comparing TOM$^-$DC2 and TOM$^+$ cDC2 groups), one-way ANOVA (comparing 2-week-old and adult groups) or two-tailed $t$-test in (**c, d, h**). Only statistically significant comparisons are indicated. Source data are provided as a Source Data file.

populations took up fluorescently labeled beads with similar efficiency (Supplementary Fig. 6H). Bead uptake was mediated by phagocytosis, as it was inhibited by cytochalasin D (Supplementary Fig. 6H). Thus, TOM$^+$ cDC2 and TOM$^-$ DC2 do not differ in their phagocytic ability. We next cultured TOM$^+$ cDC2 and TOM$^-$ DC2 from young mice and adult TOM$^+$ cDC2 with CellTrace Violet (CTV)-labeled OT-II T cells in the presence of OVA under Th17 conditions (Fig. 5i). As observed before, TOM$^+$ cDC2 and TOM$^-$ DC2 from young mice induced a higher frequency of IL-17A-producing T cells than adult cDC2 (Fig. 5i). Thus, TOM$^+$ cDC2 and TOM$^-$ DC2 from young mice are comparable to their adult counterparts in their ability to process antigen and induce naive T-cell proliferation in an antigen-specific manner. TOM$^+$ cDC2 and TOM$^-$ DC2 from young mice promoted similar Th17 and Treg differentiation; however, compared to adult cDC2, Th17 and Treg differentiation was increased, supporting the notion that distinct environmental cues in young and adult mice shape the ability of DCs to induce T-cell differentiation. Although such cytokines could be regulated in response to the microbiota, cDC2 from adult mice housed in germ-free (GF) and specific pathogen-free (SPF) conditions induced similar OT-II cell differentiation under Th0, Th17, and Treg conditions, and did not recapitulate the phenotype of young cDC2 in terms of PDL-1 and CD38 expression (Supplementary Fig. 6I, J). These data suggest that functional differences of cDC2 in early life are not simply due to a lower microbial load at this age.

**Early-life cDC2 induce distinct T-cell responses upon targeted antigen delivery compared to adult-life cDC2.** Having established that cDC2 from young mice can activate T cells in vitro, we next asked whether cDC2 from young mice have this capacity upon direct delivery of antigens in vivo. We relied on an established method to target the model antigen OVA to cDC2 by coupling it to an antibody directed against the C-type lectin receptor CLEC4A4/DCIR2[25,107]. As TOM$^+$ cDC2 and TOM$^-$ DC2 were phenotypically identical, we were unable to target either cell population individually but CLEC4A4 was expressed by both TOM$^+$ cDC2 and TOM$^-$ DC2 (Fig. 3a). We injected 2-week-old Clec9a$^{cre/+}$Rosa$^{TOM}$ mice with anti-DCIR2-OVA or isotype-matched control antibody of irrelevant specificity. Twelve hours later, we sort-purified splenic TOM$^+$ cDC2 and TOM$^-$

DC2, and co-cultured them with CTV-labeled OT-II T cells from adult mice (Fig. 6a) to overcome any T-cell intrinsic differentiation bias in early life[2]. In this experimental set up, TOM$^+$ cDC2 and TOM$^-$ DC2 induced OT-II proliferation upon targeting with anti-DCIR2-OVA but not isotype-matched control antibody (Fig. 6a). Thus, TOM$^+$ cDC2 and TOM$^-$ DC2 from 2-week-old mice could be targeted with anti-DCIR2 and process antibody–antigen complexes for presentation to CD4$^+$ T cells, although OT-II proliferation in the absence of adjuvant was low, as expected[107].

Immune responses in early life are Th2 biased but such predisposition can be overcome through the use of Th1 adjuvants, such as CpG-B[108,109]. CpG-B signals through TLR9, which was similarly expressed between the profiled cDC2 populations in bulk mRNA sequencing (Supplementary Fig. 6A). Nonetheless, TOM$^+$ cDC2 and TOM$^-$ DC2 from 2-week-old and TOM$^+$ cDC2 from adult mice showed a distinct cytokine profile after CpG-B stimulation. In response to CpG-B TOM$^-$ DC2 from young mice were the most efficient cytokine producers and secreted higher amounts of IL-6, IL-12p40, and TNF-α than cDC2 from adults (Fig. 6b). Interestingly, IL-6 and TNF-α production were significantly higher in TOM$^-$ DC2 compared to TOM$^+$ cDC2 from young mice, raising the possibility that these cells have distinct inflammatory potential. Independent of origin cDC2 from young mice produced higher IL-10 and IL-27 than their adult counterparts (Fig. 6b and Supplementary Fig. 7A). Although IL-27 production was low and not detectable in all samples, it is noteworthy, because DCs from peripheral blood of children show increased IL-27 production compared to DCs from adults[110]. Thus, despite exhibiting similar Tlr9 expression cDC2 from young mice have different cytokine response to CpG-B than cDC2 from adults, indicating distinct signaling downstream of TLR9 and supporting the hypothesis that age imprints cDC2 with distinct immune reactivity.

We next tested whether the distinct response to CpG-B stimulation would influence the ability of cDC2 to activate T cells and induce their differentiation. We injected 2.5-week-old or adult mice with anti-DCIR2-OVA in the presence of CpG-B as adjuvant. Twelve hours later, we sorted TOM$^+$ cDC2 and TOM$^-$ DC2 from young mice, as well as TOM$^+$ cDC2 from adult mice and co-cultured them with CTV-labeled OT-II T cells isolated from adult mice (Fig. 6c, d). After 3.5 days, T-cell proliferation, cytokine production, and Foxp3 expression were analyzed. In this

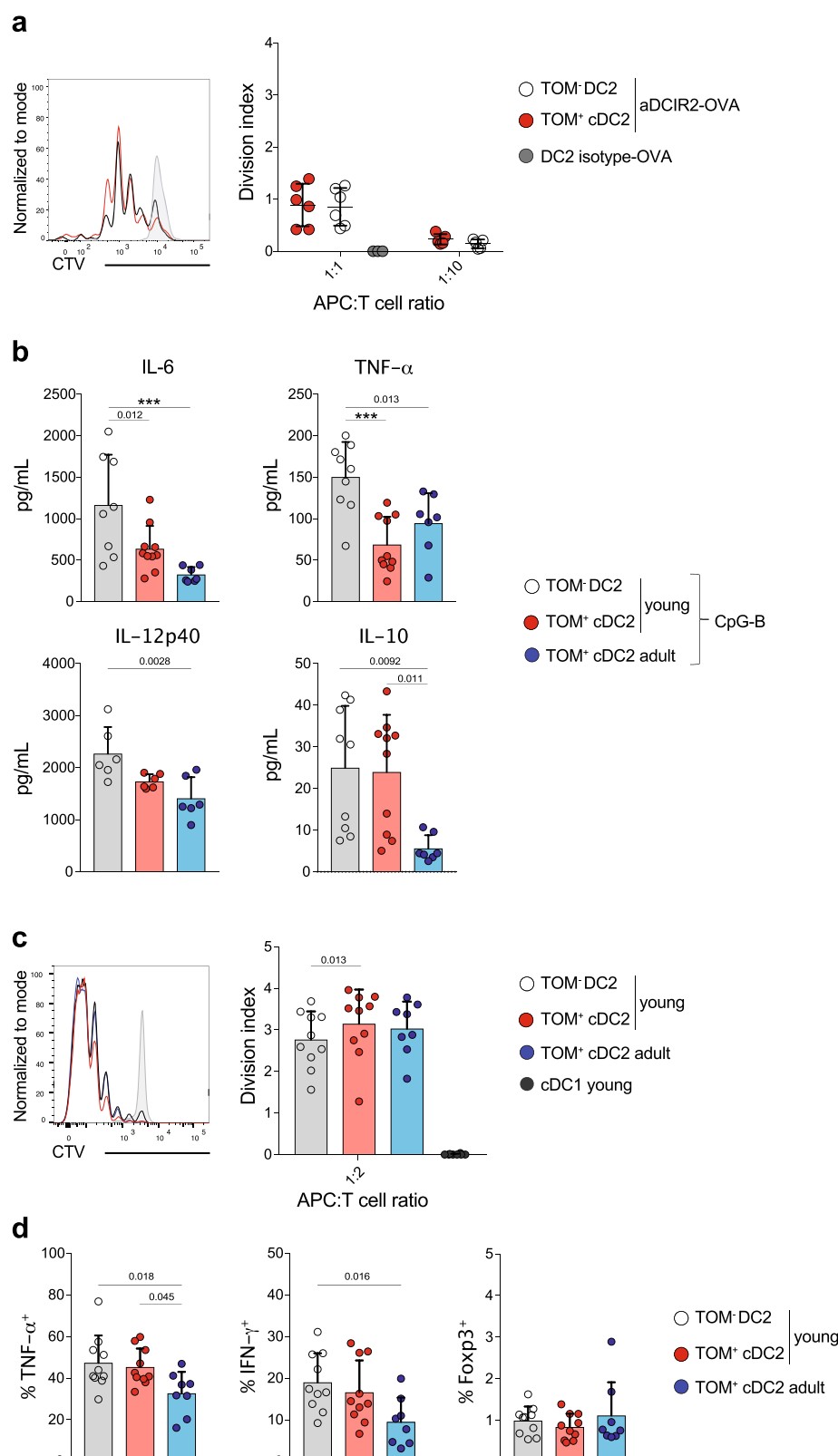

assay, TOM⁺ cDC2 and TOM⁻ DC2 from young mice and cDC2 from adult mice induced similar OT-II T-cell proliferation (Fig. 6c). Targeting was cell-type specific as cDC1 did not promote T-cell proliferation (Fig. 6c). Importantly, T cells stimulated with TOM⁺ cDC2 and TOM⁻ DC2 from young mice produced more TNF-α than T cells stimulated with cDC2 from adult mice (Fig. 6d). Notably, this is in contrast to splenic

human cDC2 from fetal tissue, which suppress T-cell-derived TNF-α through ARGINASE 2[43]. T cells stimulated with TOM⁻ DC2 from young mice also produced more IFN-γ than T cells stimulated with cDC2 from adult mice (Fig. 6d and Supplementary Fig. 7B). Increased IFN-γ production was also observed for T cells stimulated with TOM⁺ cDC2 from young mice compared to cDC2 from adults (Fig. 6d and Supplementary Fig. 7B),

**Fig. 6 Early-life cDC2 induce distinct T-cell responses upon targeted antigen delivery compared to adult-life cDC2. a** 2-Week-old *Clec9a^cre/+Rosa^TOM* mice were injected i.p. with aDCIR2-OVA or isotype-OVA control antibody. After 12 h, TOM⁺ cDC2 and TOM⁻ DC2 were sorted and co-cultured with OT-II cells for 3.5 days. CTV dilution (left) and division index of OT-II cells after co-culture with the indicated DC2 populations (right) are shown (*n* = 6, 1:1 ratio; *n* = 5, 1:10 ratio; *n* = 3, isotype-OVA). **b** TOM⁺ cDC2 and TOM⁻ DC2 from 2-week-old and TOM⁺ cDC2 from adult *Clec9a^cre/+Rosa^TOM* mice were sorted and stimulated with CpG-B. Cytokine production was analyzed 18−20 h later. Each dot represents a biological replicate from 4 (IL-6, TNF-α, and IL-10) or two (IL-12p40) independent experiments. **c, d** 2-Week-old and adult *Clec9a^cre/+Rosa^TOM* mice were injected i.p with anti-DCIR2-OVA antibody plus CpG-B. After 12 h DC populations were sorted and co-cultured with OT-II T cells as in **a. c** CTV dilution (left) and division index (right) of OT-II cells after co-culture with the indicated DC populations (*n* = 10, 2-week-old; *n* = 8, adult and cDC1 from 2-week-old mice). **d** Percentage of TNF-α, IFN-γ, and Foxp3-positive cells within proliferated OT-II cells (*n* = 10, 2-week-old; *n* = 8, adult). Each dot represents one biological replicate from at least three independent experiments, horizontal bars represent mean, error bars represent SD. ***p < 0.001. Statistical analysis was performed using two-tailed paired *t*-test (comparing TOM⁻DC2 and TOM⁺ cDC2 groups) or one-way ANOVA (comparing 2-week-old and adult groups). Only statistically significant comparisons are indicated. Source data are provided as a Source Data file.

although differences did not reach statistical significance. Thus, TOM⁺ cDC2 and TOM⁻ DC2 from young mice induced more TNF-α production from T cells and greater Th1 differentiation than adult cDC2. Increased Th1 differentiation could be related to the slight increase in IL-12p40 production in cDC2 from young compared to adult mice upon CpG-B stimulation (Fig. 6b); however, it is likely that other factors, such as costimulatory molecules (Fig. 5a), also contribute. Notably, production of Th2 cytokines and Foxp3 expression was similar between T cells stimulated with cDC2 from young and adult mice (Fig. 6d and Supplementary Fig. 7B) and Th17 cytokine production was not observed. Taken together, early-life TOM⁺ cDC2 and TOM⁻ DC2 are fully capable of inducing T-cell proliferation upon targeted antigen delivery in vivo but in the context of CpG-B stimulation early-life cDC2 induce an increased Th1 response compared to adult-life cDC2.

## Discussion

The cDC compartment in early, compared to adult life, exhibits quantitative and qualitative differences in its ability to induce immunity, leading some to suggest that cDCs in early life may be functionally immature. Here we set out to define the reasons underlying age-dependent functional differences of cDCs. We demonstrate that early-life cDC2 exhibit a distinct hematopoietic origin and, similar to other myeloid and lymphoid cells, develop sequentially at specified times during development. In contrast to other cell types that develop in waves[42,80–82], cell origin appeared negligible for cell function, as ontogenetically distinct cDC2 in early life were transcriptionally and functionally similar and within 2 weeks after birth could prime naive T cells at levels comparable to cDC2 from adult mice. Instead our data suggest that cytokine environments in early and adult life imprint cDC2 with a distinct gene expression profile that alters their ability to respond to pathogenic stimuli, secrete cytokines, and induce T-cell differentiation. Thus, cDC2 in early life are functionally capable but transcriptionally pre-disposed to respond differently to pathogens and induce distinct T-cell responses than their adult counterparts.

In early life, the majority of cDC2 originated from *Clec9a*-negative progenitors denoted here as TOM⁻ DC2. These cells were replaced within the first few weeks after birth by cells arising from bona fide *Clec9a*-expressing myeloid cDC progenitors, denoted as TOM⁺ cDC2. Coinciding with the increase in *Clec9a*-cre expression history, we found evidence of a lymphoid contribution to the early-life cDC2 compartment using *Rag1^cre* and *Il7r^cre* mice that declined with age. *Rag1^cre* also labels the progeny of yolk sac-derived lymphomyeloid primed progenitors[66] and *Il7r^cre* expression history is found in macrophages of yolk sac origin[111]. As we exclude a yolk sac origin but demonstrate cDCs to be *Myb*-dependent, our data indicate an HSC-

derived lymphoid contribution to early-life cDCs. Such lymphoid progenitors likely arise from fetal liver-resident progenitors, as we find splenic cDC2 as early as E16.5 (Fig. 2c and Supplementary Fig. 1D), at which stage hematopoiesis has yet to colonize bone marrow[112,113]. This notion is further supported by in utero *Cxcr4^creER* fate mapping[77]. A lymphoid path to generating cDCs has been suggested[69,71–74] but does not contribute to the steady-state cDC pool in adult mice[48,74,76]. Our data suggest that lymphopoiesis contributes to the cDC compartment of mice in early life. Such layered ontogeny is reminiscent of other immune cells that develop in waves during development, including macrophages, mast cells, ILCs, B, and T cells[42,80,81,114,115]. Although layered immune development can generate progressively adapted immune populations[80–82], TOM⁺ cDC2 and TOM⁻ DC2 exhibit apparently identical phenotype, transcriptional profile, and similar cell function, which is consistent with other studies comparing human and murine cDCs generated from lymphoid or myeloid progenitors[74,116]. It is possible that functional differences exist between TOM⁺ cDC2 and TOM⁻ DC2 in situations that we have not yet explored; however, lymphopoiesis may compensate DC poiesis when bona fide cDC progenitors are not present in sufficient quantities to generate a full repertoire of cDCs[74]. This hypothesis is consistent with the low frequency of CDPs and pre-cDCs in bone marrow from neonatal compared to adult mice. Whether cDC-like cells in early life arise from lymphoid-committed progenitors, such as CLPs, or from conversion of already differentiated lymphoid cells, such as pDCs or ILCs[117–119] needs to be investigated. We and others have proposed that cDCs and other mononuclear phagocytes should predominantly be defined on the basis of their ontogeny[5,114,120]; however, our data imply that an ontogenetic view to cell definition may not be plausible if populations with different origins turn out to be identical by all measures examined[74,79,121–123].

In continuation of earlier studies[9,10,17], we profiled the cDC compartment by flow cytometry with age using large marker panels. We demonstrate that the major subsets of splenic cDCs, including cDC1, non-canonical CD8⁺ cDCs, as well as ESAM^high and ESAM^low cDC2, are present in murine spleen as early as 1 week after birth. These findings were further substantiated using unbiased dissection of the cDC compartment of 1-week-old mice using scRNA-seq. In this analysis, cDC1 segregated across several clusters, possibly reflecting functional heterogeneity within this compartment[17,124]. Non-canonical CD8⁺ cDCs were positioned more closely to cDC2 than pDCs or cDC1, which is consistent with bulk transcriptome profiling in adult mice[59,60]. Although non-canonical CD8⁺ cDCs were recently suggested to constitute a transitional cell type with pDC and cDC2 characteristics in adult mice[125], our data indicate non-canonical CD8⁺ cDCs as a unique DC subset with close resemblance to cDC2 in early life. In scRNA-seq, early-life cDC2 spread across two main clusters resembling ESAM^high and ESAM^low subsets found in adult

spleen[62], which, despite being profiled at a different age, also resembled the T-bet-positive and T-bet-negative cDC2, respectively[92]. We also identified a cluster of cells resembling migratory cDCs, which has recently been observed in adult spleen[92]. Importantly, scRNA-seq revealed TOM$^-$ DC2 and TOM$^+$ cDC2 as transcriptionally identical. Although bulk RNA-seq identified differences in gene expression between TOM$^-$ DC2 from TOM$^+$ cDC2, we could correlate these genes to a unique cluster of *Rorc*-expressing cells that transcriptionally resembles cDC2 but lacks *Clec9a*$^{cre}$ expression history. Accordingly, RORγt$^+$ cells could be identified by flow cytometry within TOM$^-$ DC2 in early life but RORγt$^+$ cDC2 were not found in adult spleen. *Rorc*$^{cre}$-mediated fate mapping and trajectory analyses indicate that these cells are unlikely to serve as progenitors for cDC2 or other cDC subsets. The fact that *Rorc*-expressing cluster 11 segregated away from the putative RORγt-regulated cDC2B[92] (cluster 3) further supports that RORγt$^+$ TOM$^-$ DC2 constitute a unique cDC subtype present transiently during early life.

Although in adult spleen ESAM$^{high}$ cells constituted the dominant cDC2 population, this subset was a minority of cDC2 in mice under 1 week of age. ESAM$^{high}$ and ESAM$^{low}$ cDC2 can activate CD4$^+$ T cells but they may differ in their ability to promote effector differentiation[62,92,126]. We mostly characterized bulk cDC2 in 2- to 2.5-week-old mice when ESAM$^{high}$ and ESAM$^{low}$ subset distribution had reached adult levels, because sufficient TOM$^+$ cDC2 and TOM$^-$ DC2 could be isolated to probe cell function. At this time, splenic TOM$^+$ cDC2 and TOM$^-$ DC2 were fully capable of initiating T-cell activation but they induced greater Th17 and Treg differentiation under in vitro polarizing conditions than adult cDC2. These observations could be correlated to the unique gene expression profile of cDC2 in early and adult life. Early-life cDC2 expressed higher levels of factors known to promote Tregs, such as *Lgals1*, *Aldh2*, and *Sema4a*[100–103,106], suggesting increased tolerogenic potential of cDC2 in early life, which is also consistent with their greater IL-10 production in response to CpG-B stimulation. Higher levels of *Il6ra* on cDC2 from young mice could increase IL-6 transpresentation[104] and promote Th17 differentiation in conjunction with reduced expression of *Tnfsf4* (OX40L) and *Cd274* (PD-L1) in early life, which can suppress Th17 differentiation[97,98,127,128]. Some of these age-dependent transcriptional changes could be related to small differences in the composition of cDC2 in 1-week-old and adult mice. We nonetheless consider age to be the most prominent contributor to variation, because we demonstrated that both ESAM$^{high}$ and ESAM$^{low}$ cDC2 function differently in early and adult life.

The notion that cDC2 are equipped with an age-dependent gene expression profile that alters their ability to induce T-cell differentiation was further supported using targeted antigen delivery to cDC2 in early and adult life followed by ex vivo antigen presentation[25,107]. This approach allowed us to (a) compare cell function in vivo across age, while excluding age-dependent functional regulation of other immune components as confounding factor and (b) to demonstrate the potential to target cDC2 to induce T-cell responses. CpG-B was chosen as adjuvant, because despite similar *Tlr9* expression across age, TOM$^+$ cDC2, and TOM$^-$ DC2 from 2-week-old mice showed distinct cytokine production after CpG-B stimulation in vitro, including higher levels of IL-12p40, IL-6, IL-10, and IL-27, supporting a distinct immune reactivity of cDC2 with age. IL-27 is a pleiotropic cytokine with pro- and anti-inflammatory properties in multiple cell types[129], but after vaccination in mice its expression correlates with protective CD8$^+$ T-cell responses[130]. DCs from peripheral blood of children show higher IL-27 production than DCs from adults[110] but whether cDC2-derived IL-27 plays a role in early-life immunity needs to be investigated. Despite producing

increased amounts of IL-10, which dampens CD4$^+$ T-cell responses[131], early-life TOM$^+$ cDC2 and TOM$^-$ DC2 induced similar OT-II T-cell proliferation compared to cDC2 from adults upon DCIR2-mediated antigen targeting in the context of CpG stimulation. However, early-life cDC2 induced greater Th1 differentiation and TNF-α production from T cells. Human fetal DC2 suppress TNF-α production from T cells[43], suggesting that cDC2 function changes dynamically after birth. Transcriptional profiling of fetal and neonatal cDC2, possibly at the single-cell level, may provide further insights into the processes involved in shaping early-life cDC2 function. Differences in cytokine production and T-cell stimulatory capacity between cDC2 from young and adult mice were not simply regulated at the level of PRR expression but most probably involve downstream signaling events, costimulatory molecules, or epigenetic modification. Therefore, it is likely that age-dependent differences in cDC2 function will be magnified when cells are stimulated through PRRs with differential gene expression across age and additional adjuvants must be tested for their ability to promote cDC2-mediated T-cell priming and differentiation with age.

Our studies indicate that splenic cDC2 do not exhibit a Th2 bias, which is in contrast to cDC2 in the developing lung. In this organ, IL-33 induces OX40L expression on cDC2, which promotes Th2 skewing[30]. Therefore, site-specific cytokine environments appear to play a major role in imprinting cDC function in different organs and with age. This is consistent with previous studies[38,92] and our observation that cDC2 from adult compared to young mice showed an enrichment of genes involved in signaling downstream of IL-2, IFN-γ, TNF-α, and IFN-α. In our hands, cDC2 from GF mice did not recapitulate the phenotypic and functional features of neonatal cDC2. Although GF mice have a complete absence of commensals, in an SPF environment cDC2 are expected to encounter a variety of microbial signals within the first days of birth. We therefore believe that these observations do not exclude the possibility that microbially derived signals or signals induced in response to commensals play a role in imprinting cDC2 function with age. During weaning, the intestinal microbiota induces a cytokine response termed the "weaning reaction," which is required for normal immune development[132]. As cDC2 function in our hands has been assessed before this period, it is intriguing to speculate that cytokines or other factors that change in response to alterations in commensals or diet during weaning are involved in shaping cDC function with age. Such effects could even be mediated by lasting alterations on progenitor populations similar to the concept of trained immunity.

Neonates are highly susceptible to infections with *Candida albicans*, *Bordetella pertussis*, and *Streptococcus pneumoniae*, which can cause life-threatening complications in early life[1,133,134]. Defense against these pathogens requires Th17 responses[135,136]. Our demonstration that early-life cDC2 are fully capable of activating naive CD4$^+$ T cells, and supporting Th1 and Th17 differentiation highlights the potential of harnessing these cells for boosting protective immunity in early life. This could for instance be achieved by delivering antigens specifically to cDC2 in the context of adjuvants that target age-specific parameters of cDC2 function.

## Methods

**Mice.** Clec9a$^{tm2.1(icre)Crs}$ (*Clec9a-cre*)[48] (Jackson Laboratory Stock No: 025523), Gt (ROSA)26Sor$^{tm1(EYFP)Cos}$ (*Rosa26$^{lox-STOP-lox-EYFP}$*)[137] (Jackson Laboratory Stock No: 006148), Gt(ROSA)26Sor$^{tm9(CAG-tdTomato)Hze}$ (*Rosa26$^{lox-STOP-lox-tdtomato}$*)[91] (Jackson Laboratory Stock No: 007909), Tg(Csf1r-cre/Esr1*)1Jwp (*Csf1r$^{Mer-iCre-Mer}$*)[138] (Jackson Laboratory Stock No: 019098), Flt3l$^{tm1Imx}$ (*Flt3l$^{-/-}$*)[65] (MMRRC Stock No: 37395-JAX), Myb$^{tm1Ssp}$ (*Myb$^{+/-}$*)[68] (Jackson Laboratory Stock No: 004757), Rag1$^{tm1(cre)Thr}$ (*Rag1$^{cre}$*)[139], Tg(TcraTcrb)425Cbn (OT-II)[140] (Jackson Laboratory Stock No: 004194) crossed to a Thy1.1 (CD90.1) background, Tg(Rorc-EGFP)1Ebe

(Rorc[eGFP])[141], Cxcr4[creER77], Gt(ROSA)26Sor[tm4(ACTB-tdTomato,-EGFP)Luo]/
J (Rosa[mTmG])[142] (Jackson Laboratory Stock No: 007576), Ly5.1 (CD45.1) B6.SJL
(Jackson Laboratory Stock No: 002014), and C57BL/6J mice were bred at the Bio-
medical Center or Walter-Brendel-Centre for Experimental Medicine. Tg(Rorc-cre)
1Litt (Rorc-cre)[94] (Jackson Laboratory Stock No: 022791), Il7r[tm1.1(icre)Hrr] (Il7r[cre])[76],
and Gt(ROSA)26Sor[tm1Hjf] (Rosa26[lox-STOP-lox-RFP])[143] (EMMA strain ID EM:02112)
mice were bred at the University of Birmingham, UK. Ifnar1[tm1Agt] (Ifnar[−/−])[144]
(MMRRC Stock No: 32045-JAX) mice were bred at the University Hospital Erlangen,
Germany. Mice were maintained in SPF conditions with a 12 h dark/light cycle, in
individually vented cages (type II long, measuring $18 \times 30 \times 13$ cm with stocking
density according to EU guideline 2010/63) supplied with autoclaved bedding, play
tunnels, nestlets, and mouse houses. Irradiated food and sterile filtered and ultraviolet-
light exposed water were provided ad libitum. Cage manipulations took place in
laminar flow hoods. Air temperature was $22 \pm 2\,°C$ and humidity $55 \pm 10\%$ with daily
control and record. GF mice were provided by Dirk Haller. For timed matings, mice
of desired genotypes were mated overnight. Embryonic development was estimated
considering the day of vaginal plug formation as E0.5. Mice over the age of 8 weeks
were considered adults. The neonatal period was defined as the first 10 days of life.
Mice younger than 3 weeks of age were killed by decapitation. Adult mice were killed
by cervical dislocation. Mice were sex-matched but male and female mice were used
for all experiments. In most experiments, littermates were used. When littermates
could not be used, e.g., in experiments involving mice of different age, mice were kept
under the same barrier conditions and in the same racks. All animal procedures were
performed in accordance with national and institutional guidelines for animal welfare
and approved by the Regierung of Oberbayern.

**Cell isolation for flow cytometry.** Spleens were minced into small pieces and
digested in 1 mL of RPMI (Thermo Fisher Scientific) with 200 U/mL collagenase IV
(Worthington) and 0.2 mg/mL DNAse I (Roche) for 30 min at 37 °C while shaking.
After digestion, cells were passed through a 70 μm strainer and washed once with
FACS buffer (phosphate-buffered saline (PBS), 1% final calf serum (FCS), 2.5 mM
EDTA, 0.02% sodium azide). Erythrocytes were lysed with Red Blood Cell Lysing
Buffer Hybri-Max (Sigma-Aldrich) for 2 min at room temperature (RT), washed
once, and resuspended in FACS buffer for further analysis. Bone marrow from
adult mice was isolated from femurs and tibiae by flushing, and bone marrow from
mice under 2 weeks of age was isolated by crushing the bones through a 70 μm cell
strainer. Erythrocytes were lysed as above and cells were resuspended in FACS
buffer for further analysis. Liver was minced into small pieces and digested in 2 mL
PBS containing $Mg^{2+}$ and $Ca^{2+}$ (Sigma-Aldrich) with 1 mg/mL collagenase IV
(Worthington), 60 U/mL DNAse I (Roche), 2.4 mg/mL Dispase II (Roche), and 3%
FCS (Sigma-Aldrich) for 30 min at 37 °C while shaking. After digestion, cells were
passed through a 100 μm strainer and centrifuged for 3 min at 50 g at 4 °C, to pellet
hepatocytes. The supernatant was collected and recentrifuged for 7 min at $320 \times g$
at 4 °C. Pelleted cells were resuspended in FACS buffer for further analysis.

**Cell isolation for cell sorting and functional analyses.** Cell isolation from spleen
was performed as above but FACS buffer without sodium azide was used for all
functional and RNA profiling experiments. For sorting of cDC2 CD11c[+] cells were
enriched from splenic single-cell suspensions by positive selection using anti-
CD11c magnetic beads and LS columns (Miltenyi) according to the manufacturer's
instructions. For OT-II cell isolation, the spleen was mechanically disrupted
through a 70 μm strainer and washed once with FACS buffer without sodium azide.
OT-II cells were then enriched from total splenocytes using the EasySep™ Mouse
Naive CD4[+] T Cell Isolation Kit (Stemcell Technologies) according to the man-
ufacturer's instructions. Erythrocytes were not lysed prior to OT-II cell enrichment,
following the manufacturer's recommendations.

**Flow cytometry.** For staining of surface epitopes, cells were incubated first in 50 μL
with purified anti-mouse CD16/32 (FcBlock) for 10 min at 4 °C, before additional
antibodies were added to a final staining volume of 100 μL in a 2× Mastermix. Cells
were stained at 4 °C for 20 min then washed twice and resuspended in FACS buffer
for analysis. CCR6 staining was performed at 37 °C for 45 min before staining with
additional antibodies. For intracellular staining, cells were first stained with anti-
bodies against surface epitopes in the presence of fixable viability dye eFluor™ 780
(Thermo Fisher Scientific) and then washed with FACS buffer. Intracellular
staining for cytokines was performed using intracellular Fixation & Permeabili-
zation Buffer Set and intranuclear staining was performed with the Foxp3 tran-
scription factor staining set (both Thermo Fisher Scientific) according to the
manufacturer's instructions. A pre-fixation step with 2% paraformaldehyde at RT
for 15 min was performed after surface epitope staining to preserve TOMATO
signal during intranuclear staining. Dead cells were excluded from analysis by 4′,6-
diamidino-2-phenylindole staining (Sigma-Aldrich) for live samples or fixable
viability dye eFluor™ 780 (Thermo Fisher Scientific). Data were collected on an LSR
Fortessa (BD Biosciences) using BD FACSDiva Software (BD BioSciences, Version
8) and data analysis was performed using FlowJo software (Tree Star, Inc.). Cell
sorting was performed on an Aria III Fusion (BD Biosciences). Cells counts were
quantified using CountBright™ Absolute Counting Beads (Thermo Fisher Scien-
tific). Mean fluorescence intensity was calculated as the geometric mean of the

indicated fluorescent parameter using FlowJo software (Tree Star, Inc.). Antibodies
used for flow cytometry are provided in Supplementary Data 4.

**Immunofluorescence microscopy.** Spleens were fixed overnight at 4 °C in par-
aformaldehyde, then dehydrated in P-buffer (0.2 M $Na_2HPO_4$ and 0.2 M $NaH_2PO_4$
at an 81 : 19 analogy), containing 30% sucrose overnight at 4 °C as described[145],
transferred to Tissue-Tek O.C.T. (Sakura), and frozen on dry ice. Twelve-
micrometer-thick frozen sections were cut on a cryostat at −20 °C (Leica
CM3050S), rehydrated in PBS, and permeabilized with Acetone (Sigma-Aldrich).
Afterwards, the sections were circled with a PAP Pen (Kisker Biotech GmbH) and
blocked for 1 h at RT in the dark with blocking buffer containing 10% goat serum
in PBS. Antibodies were diluted in blocking buffer and sections were incubated for
2 h at RT in the dark with the antibody mixture. Finally, stained sections were
washed with PBS, mounted with ProLong™ Diamond Antifade Mountant (Thermo
Fisher Scientific), cured at RT for 24 h in the dark, and stored at 4 °C until imaging.
Confocal microscopy was performed at the Core Facility Bioimaging of the Bio-
medical Center with an upright Leica SP8X WLL microscope, equipped with
405 nm laser, WLL2 laser (470 - 670 nm), and acusto-optical beam splitter. Three-
dimensional tile scans were acquired with a $20 \times 0.75$ objective, image voxel size
was 180 nm in $x/y$ direction and 0.5 in $z$ direction. The following channel settings
were used: BV421 (excitation 405 nm; emission 415–470 nm), AF488 (500;
510–542), TOMATO (553; 563–591), AF594 (592; 605–640), and AF647 (650;
656–718). Recording was done sequentially to avoid bleed-through. BV421, AF488,
AF594, AF647, and TOMATO were recorded with hybrid photo detectors. Tile
scans were merged in LAS X (Leica, Version 3.4.1.17670). Images were imported in
Fiji[146] to create maximum projections, adjust brightness/contrast, and to add scale
bars. Antibodies used for microscopy are provided in Supplementary Data 4.

**Pulse labeling of yolk sac progenitors.** For labeling of yolk sac-derived macro-
phages, heterozygous Csf1r[Mer-iCre-Mer] mice were crossed with homozygous
Rosa[YFP] reporter mice. Pregnant females were injected at E8.5 with a single dose of
75 μg per gram body weight 4-hydroxytamoxifen (4′OHT, Sigma-Aldrich) sup-
plemented with 37.5 μg per gram body weight progesterone (Sigma-Aldrich)[41]. The
spleen and liver of F1 mice were analyzed at embryonic day 18.5, as well as at 2 and
4 weeks after birth by flow cytometry.

**RNA isolation, library construction, and RNA-seq analysis.** CD11c[+] cells were
enriched from splenocytes of adult and 1-week-old Clec9a[cre/cre]Rosa[TOM] mice. cDC2
were identified as live, single, autofluorescence-negative, CD11c[+]MHCII[+]CD11b[+]
cells, and divided based on TOMATO expression into TOM[+] cDC2 and TOM[−]
DC2. Spleens from 1-week-old mice were pooled to increase the yield of sorted cells.
Total RNA was isolated using column-based PicoPure™ RNA Isolation Kit (Thermo
Fisher Scientific). RNA quality was assessed using a 2100 Bioanalyzer (Agilent) and
samples with RNA Integrity Number > 8 were used for cDNA synthesis by using
ultra-low input RNA SMART-seq v4 kit (Clontech) according to the manufacturer's
instructions. cDNA was transferred to AFA Fiber Pre-Slit Snap-Cap $6 \times 16$ mm
microTUBEs (Covaris) and sheared by sonication. Sheared cDNA was cleaned using
ethanol precipitation and sonication efficiency was determined using the 2100
Bioanalyzer. A maximum of 10 ng sheared cDNA was used to generate libraries for
RNA-seq with the MicroPlex Library Preparation kit v2 (Diagenode). The libraries
were amplified until a DNA concentration above 5 ng/μL was reached as determined
by Qubit 2 DNA quantification (Thermo Fisher Scientific). Amplified libraries were
cleaned using AMPure XP beads (Beckman Coulter) as described in the SMART-seq
v4 kit (Clontech) protocol and the final concentration, as well as the purity of the
libraries were assessed by using the 2100 Bioanalyzer. Sequencing was performed
on an Illumina HiSeq1500 sequencer with 50 base pair single-end reads and a
sequencing depth of 20 million reads per sample. For analysis, RNA-seq reads were
mapped to the mouse genome (mm10) using STAR[147]. Expression of genes in
transcripts per million was calculated with RSEM[148]. RNA-seq analysis was per-
formed in R (Version 3.5.3) with R-Studio (R-Studio, Inc., Version 1.1.383). Differ-
ential gene expression analysis and PCA was performed using DESeq2 (Version
1.22.2). Genes with average gene counts < 1 were discarded and $\log_2$ FC shrinkage was
performed using the Apeglm package[149]. Heatmaps were generated using pheatmap
(Version 1.0.12) and graphs were plotted with ggplot2 (Version 3.0.2). Tables con-
taining differentially expressed genes were created using Microsoft Excel
version 16.34.

**scRNA-seq and data processing.** MHCII[+] cells from splenocytes of 9-day-old
female Clec9a[cre/cre]Rosa[TOM] mice were sorted after gating out CD19[+] B cells and
autofluorescence[+]F4/80[hi] macrophages. To increase cell yield, spleens from three
mice were pooled. Sorted cells were pelleted, resuspended at $1 \times 10^3$ cells/μL in PBS
containing 0.04% bovine serum albumin, and loaded onto the Chromium Con-
troller (10X Genomics). Samples were prepared for single-cell encapsulation and
cDNA library generation using the Chromium Single Cell 3′ v3 Reagent Kits (10X
Genomics). The constructed library was sequenced on an Illumina HiSeq2500
(Rapid Run) sequencer with 28 (read 1) + 91(read 2) base pair paired-end reads
and a sequencing depth of 320 million reads in total. Sequencing data were pro-
cessed using 10X Genomics Cell Ranger v3.0.2 pipeline. Sequencing reads were
mapped to the mouse genome (mm10) using STAR[147] after spiking in the

sequences of iCRE (GenBank ID: AY056050.1), the predicted transcript of the unrecombined *Rosa* locus and *Tomato* sequence (GenBank ID: AY678269.1). Cell Ranger's count pipeline was run under default parameters. The output of Cell Ranger's count pipeline was a gene-barcode matrix consisting of ~1788 cellular barcodes with 179,245 mean reads per barcode. This matrix was inserted to the R (Version 3.5.3) software package Seurat (v3.0.2)[86,87] for all downstream analyses. Cells expressing <200 genes, having >7.5% of mitochondrial associated genes and genes detected in <3 cells were removed from further analysis according to the software suggestions. In addition, cells were scored based on their cell cycle score and the differences between G2M and S phase cells were regressed. The SCtransform package was used to normalize, scale, and find the variable features of the dataset before PCA. Genes associated with the top 40 PCs were used for graph-based cluster identification (resolution 1.3) and subsequent dimensionality reduction using UMAP. Identification of cluster defining markers and differential expression analysis were performed using the FindAllMarkers and FindMarkers commands of the Seurat package. To score single cells based on their *Tomato* expression, we calculated the ratio of normalized *Tomato* reads per cell to normalized reads per cell of the predicted transcript of the unrecombined *Rosa* locus (Ratio Tom/unrec. Rosa). Cells with a ratio above 0.5 or below −0.5 were denoted *Tom*⁺ or *Tom*⁻, respectively. The selected thresholds included more than 90% of the total cells in each cluster. Tables containing differentially expressed genes were created using Microsoft Excel version 16.34.

For trajectory analysis using Palantir, we selected clusters 0, 3, 6, 7, 11, and 14, and generated diffusion maps as described[93]. We provided a defined starting point in cluster 11 and used Palantir to characterize potential pseudo-time trajectories from this point. The terminal states found in the analysis were automatically calculated by the Palantir algorithm.

**Cytospins and Hemacolor staining.** TOM⁺ cDC2 and TOM⁻ DC2 were sorted from CD11c⁺ enriched splenocytes from 11-day-old *Clec9a*^cre/cre^*Rosa*^TOM^ mice as live, single, autofluorescence-negative, CD11c⁺MHCII⁺CD11b⁺ TOM⁺ or TOM⁻ cells. Cells (2 × 10⁴) were spun onto a microscope slide in a Shandon Cytospin 2 for 5 min at 8000 r.p.m. and stained with Hemacolor® rapid staining kit (Merck). Microscopy was performed at the Core Facility Bioimaging of the Biomedical Center using a Leica DM 2500 LED microscope with a ×100 magnification.

**Bone marrow FLT3L cultures.** Bone marrow cells from adult *Clec9a*^cre/+^*Rosa*^TOM^ (CD45.2) were isolated as described above.

After lineage depletion using negative selection (lineage: CD3, TER119, Ly6G, CD19), CLPs were sorted as live, single, lin⁻CD115⁻CD117^int^CD135⁺CD127⁺ B220⁻TOMATO⁻ cells. CLPs (10⁴) were seeded in 300 μL complete medium (RPMI supplemented with 10% FCS, penicillin/streptomycin, 1% non-essential amino acids, 1% sodium pyruvate, 1% ʟ-glutamine, 0.05 mM β-mercaptoethanol) in a 48-well plate together with 1.5 × 10⁵ total bone marrow cells isolated from congenic B6.SJL (CD45.1) mice. As controls, total bone marrow cells from *Clec9a*^cre/+^*Rosa*^TOM^ and B6.SJL congenic mice were seeded at a 1 : 1 ratio. FLT3L (purified from supernatant of CHO-flk2 cell line, a kind gift from Dr. Anne Krug, Institute for Immunology, Biomedical Center, Munich) was added to all wells at a concentration of 50 ng/mL. Culture wells were left unperturbed for 7 days and then the culture output was analyzed by flow cytometry.

**In vitro T-cell proliferation.** DCs were sorted from CD11c⁺ enriched cells from spleen of 2-week-old and adult *Clec9a*^cre/cre^*Rosa*^TOM^ mice, as live, single, autofluorescent-negative, CD11c⁺MHCII⁺CD11b⁺ TOMATO⁺, or TOMATO⁻ cells. In some experiments, DCs were further subset based on ESAM and RORγt expression, as indicated. In experiments comparing cDC2 from SPF and GF mice, cDC2 were sorted from CD11c-enriched splenocytes as live, single, autofluorescence-negative, CD11c⁺MHCII⁺CD11b⁺ cells. Sorted cells were incubated in the wells of a V-bottom 96-well plate with 10 μg/mL chicken OVA peptide (OVA₃₂₃₋₃₃₉, InvivoGen) for 3 h in complete medium (prepared as indicated above). After washing two times to remove any residual OVA₃₂₃₋₃₃₉ DCs were resuspended in complete medium, serially diluted, and co-cultured at the indicated ratios with naive CTV-labeled OT-II cells from adult mice. Naive OT-II cells were isolated as described above and labeled with CTV (Thermo Fisher Scientific). First, OT-II cells were resuspended in PBS at 20 × 10⁶ cells/mL and an equal volume of CTV working solution was added to a final concentration of 5 mM. Cells were incubated for 20 min at 37 °C in the dark and were mixed at the midpoint of the incubation. An excess of pre-warmed complete medium was added and after a 5 min incubation, cells were washed with PBS. Cultures were supplemented with 20 ng/mL IL-12 (PeproTech) and 10 μg/mL anti-IL-4 (Biolegend) for Th1; 5 ng/mL TGF-β, 10 μg/mL anti-IL-4, and 10 μg/mL anti-IFN-γ for Treg; 5 ng/mL TGF-β, 20 ng/mL IL-6, 10 μg/mL anti-IL-4, and 10 μg/mL anti-IFN-γ (all Biolegend) for Th17 conditions. After 3.5 days of culture, supernatant was collected from each well and cells were restimulated with 10 ng/mL phorbol 12-myristate 13-acetate (PMA) (Calbiochem) and 1 μg/mL ionomycin (Sigma-Aldrich) for 5 h. After 2 h, brefeldin A (5 μg/mL, Biolegend) was added for the remaining 3 h. Cytokines and Foxp3 expression were detected by intracellular staining. Cytokine secretion

in culture supernatants was quantified using LEGENDplex^TM^ Mouse Th Cytokine Panel (Biolegend) according to the manufacturer's instructions. For co-culture experiments with OVA, DCs were sorted as above resuspended in complete medium in the wells of a V-bottom 96-well plate, serially diluted and co-cultured at the indicated ratios with CTV-labeled naive OT-II cells in the presence of 20 μg/mL OVA (Hyglos). After 4 days of culture, cytokine production was assessed as above.

**Antigen targeting.** Antigen targeting was performed as described[25,107]. To establish targeting specificity 2-week-old *Clec9a*^cre/+^*Rosa*^TOM^ mice were injected intraperitoneally (i.p.) with 10 μg αDCIR2-OVA or 10 μg OVA coupled isotype control antibody. Mice were killed 12 h later and splenocytes were isolated by mechanically disrupting the spleens through a 70 μm strainer and washing once with PBS containing 1% FCS. Splenic single-cell suspensions were then enriched for CD11c⁺ cells by positive selection using anti-CD11c magnetic beads and LS columns (Miltenyi). PBS containing 1% FCS was used throughout the procedures of isolation, enrichment and preparation of cells for sorting. TOM⁺ cDC2 and TOM⁻ DC2 were sorted as CD11c⁺B220⁻Ly6G⁻Ly6C⁻CD90.2⁻CD11b⁺ TOMATO⁺ or TOMATO⁻ cells and cDC1 were sorted as CD11c^high^B220⁻Ly6G⁻Ly6C⁻CD90.2⁻CD24⁺XCR-1⁺ cells. Sorted DCs were co-cultured with CTV-labeled naive CD90.1 OT-II cells in complete medium. After 3.5 days, supernatant was collected from each well and OT-II proliferation analyzed by flow cytometry. For targeting experiments with adjuvant, 2-week-old and adult, *Clec9a*^cre/+^*Rosa*^TOM^ mice were injected i.p. with 10 μg of αDCIR2-OVA plus 0.2 μg/g body weight of CpG-B ODN 1826 (Sigma-Aldrich). DC populations were sorted and co-cultured with CTV-labeled naive CD90.1 OT-II cells as above. After 3.5 days, OT-II cells were restimulated with PMA/ionomycin for 5 h in the presence of Brefeldin A and cytokine production and Foxp3 expression assessed by flow cytometry as described above. T-cell proliferation, cytokine production after restimulation, and quantification of secreted cytokines in culture supernatants were assessed as above.

**TOMATO stability in vitro.** TOM⁺ cDC2 and TOM⁻ DC2 from 1-week-old *Clec9a*^cre/cre^*Rosa*^TOM^ mice were sorted as above. Cells (2.5 × 10³) were seeded in 200 μL complete medium together with 10⁵ total splenocytes from CD45.1 congenic mice. Cultures were supplemented with 50 ng/mL murine recombinant GM-CSF (PeproTech) and 200 ng/mL murine recombinant FLT3L (R&D Systems). After 24 h, cells were collected and analyzed by flow cytometry.

**In vitro phagocytosis.** CD11c-enriched splenocytes (5 × 10⁴) from 2-week-old and adult *Clec9a*^cre/+^*Rosa*^TOM^ mice cells were cultured with beads (FluoSpheres®, Polystyrene Microspheres, 1.0 μm, yellow–green fluorescent (505/515)), Invitrogen) at a 50 : 1 bead to cell ratio for 2 h at 37 °C in humidified atmosphere containing 5% CO₂ in complete medium. To inhibit phagocytosis, cells were pre-treated with 10 μg/ml cytochalasin D (Sigma-Aldrich) for 1 h at 37 °C. After incubation, antibody staining for surface epitopes was performed as described above and phagocytosis of beads by cDC2 was quantified by flow cytometry.

**In vitro stimulation and cytokine production.** TOM⁺ cDC2 and TOM⁻ DC2 from 2-week-old *Clec9a*^cre/+^*Rosa*^TOM^ mice and TOM⁺ cDC2 from adult *Clec9a*^cre/+^*Rosa*^TOM^ mice were sorted as above. Then, 0.7 × 10⁶ cells/mL were stimulated in a total volume of 50 μL complete medium in 96-well V-bottom plates with 0.5 μg/mL CpG-B ODN 1826 (Sigma). After 18–20 h, cytokine secretion was quantified using LEGENDplex^TM^ Mouse Inflammation Panel and LEGENDplex^TM^ Mouse Cytokine Panel 2 for IL-12p40 (both from Biolegend) according to the manufacturer's instructions.

**Statistical analysis.** Statistical significance was calculated in Prism 7 software (GraphPad) using two-tailed *t*-test with Welch's correction (unless otherwise stated) or two-tailed paired *t*-test. For multiple comparisons, one-way analysis of variance with Tukey's test (unless otherwise stated) was performed. A *p*-value < 0.05 was considered significant.

**Reporting summary.** Further information on research design is available in the Nature Research Reporting Summary linked to this article.

## Data availability
The authors declare that the data supporting the findings of this study are available within the paper and its Supplementary Files. Raw data are available from the authors upon reasonable request. Datasets related to bulk and single-cell sequencing experiments that were generated and analyzed for the current study have been deposited and made publicly available in the Gene Expression Omnibus under the accession number GSE151595. The sequences of iCRE and Tomato used for the single-cell RNA sequencing analysis are publicly available and can be found under the GenBank IDs AY056050.1 and AY678269.1, respectively. Source data are provided with this paper.

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

## Acknowledgements

We thank members of the Schraml lab, Anne Krug, Caetano Reis e Sousa, and Markus Sperandio for helpful discussions and critical reading of the manuscript. We acknowledge the Core Facility Flow Cytometry and the Core Facility Bioimaging at the Biomedical Center, Ludwig-Maximilians-Universität München, for providing equipment and expertise. High-throughput sequencing was performed by the laboratory for Functional Genome Analysis (LAFUGA) of the LMU Munich. Some flow cytometry experiments were performed at the Core Unit Cell Sorting and Immunomonitoring of the University Hospital Erlangen. We thank Hans-Reimer Rodewald for providing *Il7r^cre* mice. This work was supported by ERC Starting Grant awarded to B.S. (ERC-2016-STG-715182). Work in the Schraml lab is also funded by the Deutsche Forschungsgemeinschaft (DFG, German Research Foundation) Emmy Noether Grant: Schr 1444/1-1 (to B.S.) and Projektnummer 360372040 – SFB 1335/P08 (to B.S.). C.O. is supported by an ERC Starting Grant (ERC-2016-STG-716718) and by DFG within CRC1371 (project P07) and FOR2599 (project P07). C.S. was supported by the SFB914 (project A10), as well as the DZHK (German Centre for Cardiovascular Research) and the BMBF (German Ministry of Education and Research) (grant 81Z0600204). D.D. received funding from the DFG (DU548/5-1, CRC1181-A7) and Agency national research (ANR)/German Research Foundation program (DU548/6-1).

## Author contributions

N.E.P., N.S., S.R., K.R., J.P., V.K., C.H.K.L., R.F., J.S., D.W.G., R.M., D.M., and B.S. performed experiments. N.E.P., G.S., and T.S. performed sequencing analysis. C.H.K.L. and D.D. helped with targeting experiments. C.O. provided *Rorc^eGFP* mice, D.H. germ-free mice, S.E.J. *Rag1^cre* mice, and R.S. CXCR4^CreER mice, and helped in planning the corresponding experiments. S.E.J. and J.C. provided critical input for fate mapping lymphoid progenitors. C.S., G.S., D.D., and D.W. provided reagents and critical intellectual input. B.S. designed and supervised the study.

## Funding

## Competing interests

The authors declare no competing interests.
