## [Peer Review File · Nature Communications]

REVIEWER COMMENTS

Reviewer #1 (Fate mapping, gut immunity) (Remarks to the Author):

The submitted manuscript by Schraml and co-worker provides an analysis of classical dendritic cells (cDCs) development in young and adult mice. Using an array of assays including several transgenic models of genetic fate labelling, the authors report that neonatal cDC2, unlike their adult counterparts, arise from Clec9a negative precursors. Using scRNA-Seq analysis the authors determine functional and transcriptional differences and similarities between ontogenetically distinct cDC2 subsets. Despite the ability to prime and expand T cells in DC:T co-culture assays, these differences translate into functional distinctions between adult and neonatal cDC2 considering their response external stimulation. This report demonstrates a previously unappreciated heterogeneity within the pool of T cell priming DCs in the early weeks of life. Collectively, this report provides a very thoughtful and thorough analysis with relevant implications for future vaccine design.

We hope that our suggestions below will help to improve the quality of this report even further.

Major:

The authors demonstrate lymphoid contribution to cDC2 development in neonatal mice. Would adoptively transferred or cultured lymphoid precursors generate TOM- cDC2
The authors nicely address that yolk-sac hematopoiesis does not contribute to neonatal cDC2 development. What is the contribution of fetal liver-resident precursors to the pool of neonatal cDC2?

Does the RAG1-labelled pool of precursors in the neonatal liver or bone marrow contain MDPs or CDPs?

Would a fetal DC progenitor exclusively generate TOM-cDC2 in an adult mouse?

On several occasions the authors refer to environmental signal as regulator of cDC2 function. What is the impact of the microbiome as environmental trigger on the phenotype of cDC2? Would cDC2s isolated from adult germ-free mice share or retain functional features of a neonatal cDC2?

Considering the impact of microbial colonization on a plethora of immunological processes, would cDC2 isolated from adult type I interferon-deficient mice recapitulate the phenotype of neonatal cDC2?

The authors report distinct expression levels of TLR4 and TLR5 on cDC2 isolated at different time points of development. Could treatment of neonatal mice with LPS or Flagellin revert the "tolerogenic" phenotype of cDC2 or influence their respond to other TLR ligands in light of trained immunity? Do these signal impact the bone marrow CDP pool?

Even though TLR9 expression is reported to be similar in cDC2 from adult and young mice, cytokine responses are differing. These findings suggest functional differences between ontogenetically distinct DCs. Do TOM- cDC2 display phagocytic differences?

While we acknowledge that RORc^{-/-} mice have several developmental defects, do TOM- cDC2 require RORc for their development?

Minor:

Are the axis on the first dot plot in figure 1H are mislabeled?

Page 8, first paragraph of section: Reference to Fig 3C should instead be Fig 2C

Fig 4E: Since TOM- cDCs constitute a very small fraction of the adult splenic cDC2 pool (Fig 1F), it may be informative for the reader to include a representative gating strategy of the adult sample for this figure as a supplementary figure.

Reviewer #2 (Immune regulation, immune-stromal interaction) (Remarks to the Author):

Papaioannou et al. Age-dependent environmental signals rather than layered ontogeny imprint the function of type 2 conventional dendritic cells.

In this paper, the authors use cre-mediated fate mapping to track the progeny of clec9a-expressing dendritic cell precursors. They found that in neonatal mice, cDC2 arise from Clec9a+ and Clec9a- precursors, as opposed to adult mice in which almost all splenic cDC2 arise from Clec9+ precursors. Transcriptome analysis revealed that both neonatal cDC2 subsets greatly differ from the adult cDC2s but also that the minor variation among neonatal cDC2 subsets could be caused by proliferating cDC2. ESAM+ and ESAM- DCs are found in both tomato+ and tomato- subsets, whereas RORc-expressing DCs are found only in the tomato- subset in neonates. Ex-vivo co-culture of cDC2 with OT-II CD4 T cells and under polarizing conditions showed few functional differences among cDC2 subsets. Using anti-DCIR2 antibody to deliver OVA to cDC2, the authors showed a slightly higher capacity of neonatal cDC2s to induce IFN-g production in CD4 T cells ex-vivo.

Overall, this work aimed to define the factors that drive the phenotypical and functional differences previously reported between adult and neonatal cDC2 population. Although the authors find that tomato+ DCs increase with time, we have no idea about the separate precursors to the tomato- cells. Moreover, tomato expression is not the main distinction of function, whereas time after birth is more related to functional differences, perhaps related to functional differences between DC2 subsets that are not explored here. Thus, the authors assert that the reason behind age-dependent functional differences of cDCs is due to differences in cytokine environments that imprint cDC2 with a distinct gene expression profile, which is likely true, but they did not provide any evidence to support their statement. What cytokines? Where? In the end, we know that the frequencies of various DC populations change over time, but we don't know how or why or whether these subsets are functionally distinct at the different times.

1. A major point of the manuscript is that many of the neonatal cDC2 cells are derived from different (CLEC9-) precursors than the adult cDC2 cells, which are mostly derived from CLEC9+ precursors. Despite this difference, the resulting cDC2 cells are phenotypically and transcriptionally identical. An alternative to this surprising result is that neonatal and adult cDC 2 cells are derived from exactly the same precursors and that the only difference is the efficiency of recombination – perhaps because the cDC2 cells in neonates transition very rapidly through the precursor stage. Is tomato expression in the pre-cDC cells (gated as in Fig 1 G-H) the same in neonates and adults?
2. Although the authors claim that the neonatal cDC2 cells come from different precursors, they seem to be uniformly dependent on FLT3 and Myb, but independent of CSF1 (Fig 2) and have similar phenotype and placement (fig 3), again suggesting that tomato labeling is not really a distinguishing feature and may be an artifact.
3. The authors sorted DCs based on tomato expression and evaluated T cell responses. Most differences were related to age rather than the expression of tomato. Of course we know that the composition of cDC2 cells changes over time based on ESAM expression and that ESAM does not correlate with tomato expression, but that ESAM+ and ESAM- DCs have different transcriptional

programs and stimulate different types of T cell responses. Do ESAM+ and ESAM- cDC2 cells have different functions with age?

4. The other difference between neonatal and adult cDC2 cells is the presence of the RORc-expressing population 11 that is highest in neonatal tomato- cells and essentially not present in adult cells. These are also functionally distinct, and differ in frequency in the tomato+, tomato- and the adult cDC2 populations. The frequency of these cells changes over time – does their function change? Without accounting these differences in composition, I cannot interpret the T cell responses.

5. Figure 1H - X axis should be CD43?

6. Fig 4F – it would help the reader a great deal if this panel (or a copy of this panel) had outlines surrounding the clusters with a callout as to their identity. For example, clusters 1, 2, 4, 9 and 13 can be outlined together and labeled as cDC1 cells, whereas 0, 3, 7 and perhaps 11 can be outlined together and called out as DC2 cells.

Reviewer #3 (DC development, single cell analysis) (Remarks to the Author):

Summary:

The study of Papaioannou et al characterizes the development and lineage affiliation of conventional dendritic cells (cDCs) throughout neonatal and adult life. Using the Clec9a lineage tracer mouse, the authors show that cDC2 are initially generated from Clec9a- progenitors which are progressively replaced by Clec9a+ CDP derived cDC2s. From RNA Sequencing analysis they show that age appears being the major driver for differential gene expression as well as function within the cDC2 subsets.

Originality and significance:

The authors use the myeloid specific Clec9a Cre lineage tracer mouse model to assess the developmental origin and the possible functional differences of splenic cDC2 from early postnatal age to adulthood. It had been previously observed that fetal DCs are less capable of priming and show a Th2 bias. As no major difference could be phenotypically detected the authors perform bulk as well as single cell RNA sequencing. The major finding resides in the observation that cDC2 derived from young mice despite showing a reduced inflammatory profile are capable of inducing a stronger Th response which is polarized towards Treg and Th17.

The observation and the dataset are novel, however more experiments would be required to validate the findings.

Major Points:

The authors used the Clec9a-Cre x Rosa26LSL-TOM lineage tracer model to assess the developmental history of cDC2 in neonatal as well as adult mice. It would be essential to include a detailed analysis of the labeling efficiency across myeloid and lymphoid progenitors (MDPs, CDP, pre-DCs, CLP) across the first days/weeks of age. In particular three studies have recently shown the contribution of lymphoid progenitors to the cDCs and pDCs pool (Ref 74- Ref 112, please include Rodrigues et al.) It would be important to include in the study pDCs as those were suggested to be mostly (Ref 74/ Rodrigues et al.) or exclusively (ref. 112) lymphoid derived and would therefore validate the Clec9a+ CDP origin of adult cDC2 versus CLP or Clec9a- CDP derived Young cDC2.

The heterogeneity across the used models Tomato-Cre; Cre homozygous and YFP-Cre, should be discussed in more detail as the different ontogeny based on Clec9a is a major point of the study. Rag expression from Rag-Cre x Rosa26LSL-YFP is here used to trace lymphoid derived cDC2, how is

the the labelling distributed across ESAM+ and ESAM- cDC2 subsets?

Proliferation related genes appear to be the mostly differentially expressed genes that characterize young mice. What is the survival in culture of Adult versus young DCs. Could the better priming capacity of Young DCs be explained by higher proliferation and survival in culture?

The authors identified a Rorgt+ cDC2 subset within the Tom- fraction. A similar subset was recently described by the group of Rudensky (ref. 89). It would be relevant to include a detailed phenotypic characterization of the subset (CD11b, Clec4a, Sirpa, CD26, ESAM, CD117, CD80, MHC-II, CD24).

It would be important to clearly mention that the Bulk RNA sequencing is performed at 8 days of age and therefore as shown young as well as adult derived cDC2 subsets differ in their composition especially in the % of ESAM expressing cells, please refer to the related supplementary data. Similarly, in the T cell stimulation experiments cDCs from a 2.5 weeks of age mice are used, again it is important to point out to the exact composition of the cDC2 compartment (Figure 1E, 4 and 5). This could also be the reason for the PCA segregation of adult DCs, which would include a subset absent in young mice.

The authors elegantly overcome the problem of target a specific subset by delivering antigen and subsequently sorting individual subsets. Could the delivery of Antigen loaded DCs from Young versus Adult mice reproduce the results obtained?

However, we believe that a major problem in the T cell experiments performed in vitro by the authors is the inclusion of polarizing cytokines in the assay, which could have a dominant effect. We would therefore suggest to repeat the experiments without adding polarizing cytokines. A conclusion of the data which show that increased Treg and Th17 polarization is the consequence of increased IL6 and IL10 production would be important.

Minor Points:

There is a mistake in the gating strategy shown in Fig 1H and Suppl. Fig2D. The x axis of the second panel should be CD43 and not MHC-II.

The heatmap in Figure 4G is of very poor quality and does not provide much information. Please improve the quality of this figure by adding additional markers and increasing the quality.

Reviewer comments in black

Authors response blue.

Changes in the manuscript are indicated in green.

We thank the reviewers for their support and constructive comments. In the revised manuscript we have added new data that highlight the faithfulness of the *Clec9a-Cre* system in early life and we validated the lymphoid contribution to neonatal cDC2 using *Il7r-Cre* mice. We provide additional functional experiments that strengthen the conclusion that age-dependent differences in DC function are caused by distinct cytokines acting on DCs in early and adult life. These experiments are outlined in the point by point response below and we have also revised the text to maximize clarity. We hope that the reviewers now find the manuscript suitable for publication in Nature Communications.

Reviewer #1 (Fate mapping, gut immunity) (Remarks to the Author):
The submitted manuscript by Schraml and co-worker provides an analysis of classical dendritic cells (cDCs) development in young and adult mice. Using an array of assays including several transgenic models of genetic fate labelling, the authors report that neonatal cDC2, unlike their adult counterparts, arise from *Clec9a* negative precursors. Using scRNA-Seq analysis the authors determine functional and transcriptional differences and similarities between ontogenetically distinct cDC2 subsets. Despite the ability to prime and expand T cells in DC:T co-culture assays, these differences translate into functional distinctions between adult and neonatal cDC2 considering their response external stimulation. This report demonstrates a previously unappreciated heterogeneity within the pool of T cell priming DCs in the early weeks of life. Collectively, this report provides a very thoughtful and thorough analysis with relevant implications for future vaccine design. We hope that our suggestions below will help to improve the quality of this report even further.

Major:

1. The authors demonstrate lymphoid contribution to cDC2 development in neonatal mice. Would adoptively transferred or cultured lymphoid precursors generate TOM- cDC2

As suggested, we have cultured common lymphoid progenitors (CLPs) in FLT3L, which is a common culture system thought to generate cDC1 and cDC2 equivalent cells (Naik et al., J. Immun. 2005). These data are now included in new Suppl. Fig. 3B. We sorted CLPs ($\text{lin}^- \text{CD11b}^- \text{CD115}^- \text{CD117}^{\text{int}} \text{CD135}^+ \text{CD127}^+ \text{B220}^- \text{TOMATO}^-$) from bone marrow of adult *Clec9a*^{Cre/+}*Rosa*^{Tom} mice and used unfractionated bone marrow as control. In these experiments both total bone marrow and CLPs generated CD11c⁺MHCII⁺ cDC like cells, although the frequency of cDC2 equivalent cells generated from CLP was lower than from bone marrow control. cDC like cells generated from CLP and bone marrow contained CD24⁺ cDC1 and CD172a⁺ cDC2 equivalents, although it was noteworthy that CLP-derived cDC2 equivalents exhibited a slightly different surface phenotype with higher levels of CD24. Importantly, cDC2-like cells generated from CLPs were predominately TOMATO negative and labelled with TOMATO at a mere 1.05±0.94%. In contrast, cDC2 derived from unfractionated bone marrow labelled with Tomato at 13.6±0.98%. Tomato labelling of *in vitro* generated cDC2 equivalents is therefore lower than labelling of cDC2 in spleen from *Clec9a*^{Cre/+}*Rosa*^{Tom} mice (93±1.28% TOMATO⁺) but this is expected as Cre is often less efficient *in vitro*. Additionally, it is conceivable that CLP contribute to cDC2 generation in these conditions, which would lead to an additional dilution of the TOMATO signal. Thus, CLPs can generate cDC2 equivalents upon *in vitro* culture, but these are predominantly TOMATO negative, supporting the notion that lymphoid progenitors can generate TOM- DC2.

2. The authors nicely address that yolk-sac hematopoiesis does not contribute to neonatal cDC2 development. What is the contribution of fetal liver-resident precursors to the pool of neonatal cDC2?

The reviewer raises an interesting question. Our exclusion of a yolk sac origin of cDC2 combined with finding cDC2 in the spleen already as early as E16.5 (Fig. 2 B,C), at which stage hematopoiesis has yet to colonize the bone marrow (Christensen, et al., PLoS Biol, 2004; Hoeffel and Ginhoux, Cellular Immunology, 2018), strongly suggest that fetal liver-resident progenitors constitute a major source of cDC2 in early life. This is now discussed on page 24 of the revised manuscript.

Addressing the contribution of fetal liver-resident progenitors to the neonatal cDC2 pool experimentally is challenging, as there are currently no fate mapping models allowing specific labelling of fetal liver HSCs or more differentiated progenitors. *Flt3-Cre* mice have been used to dissect fetal liver hematopoiesis (Boyer et al., Cell. Stem. Cell., 2011; Hoeffel et al., Immunity, 2015) but are not suitable to map the fate of cDC2, as they also express Flt3 in their differentiated state. We therefore looked for alternative ways to experimentally determine the contribution of fetal liver hematopoiesis to the neonatal cDC2 pool. During E12.5 to E14.5 fetal liver resident precursors are expected to almost exclusively contribute to hematopoiesis (Hoeffel and Ginhoux, Cellular Immunology, 2018). We therefore decided to attempt to label proliferating cells *in utero* at E12.5 using intraperitoneal injection of the thymidine analog EdU into pregnant dams. EdU is expected to cross the placenta and incorporate into proliferating cells, including fetal liver progenitors, during a window of about 6 hours (Magavi and Macklis, Methods Mol. Biol., 2008). We then analyzed EdU incorporation in splenic DCs of newborn mice, which should be an indication of labelling during E12.5. Unfortunately, we were not able to detect any remaining EdU signal in splenic DCs or other lymphoid or myeloid cell types. These data have been included for the reviewer in Figure 1A below. EdU signal was detected in the splenic DCs of the mother, suggesting that the EdU injection was technically successful (Figure 1A below). Although thymidine analogues have been successfully used to label cells *in utero* (Magavi and Macklis, Methods Mol. Biol., 2008; Pulvers and Huttner, Development, 2009), it is possible that the conditions used in our hands were not optimal and that EdU did not cross the placenta. Alternatively, it is possible that progenitors were labelled but due to active cell division the EdU signal diluted to non-detectable levels during the time frame analyzed. We concluded that without extensive optimization of the EdU labelling conditions and timing of analysis, this approach would not allow to determine the contribution of fetal liver hematopoiesis to neonatal cDC2.

It was previously shown that fetal liver MDPs, the immediate progenitors of fetal monocytes express *Cx3cr1* (Hoeffel et al., Immunity, 2015). In an attempt to fate map these fetal liver myeloid progenitors, we crossed *Cx3cr1-CRE-ERT2* mice to *Rosa^{Tom}* mice and injected pregnant dams with tamoxifen injection on E12.5. This strategy should allow for a continuous labelling of progenitors over the course of a few consecutive embryonic days. TOMATO labelling was assessed in offspring mice on E21.5. We reasoned that this approach would allow to exclude a contribution of fetal liver monocytes to neonatal cDC2 and thus strengthen the conclusion that neonatal cDC2 have a lymphoid origin. Microglia, which express *Cx3cr1* and at the time of labeling are fully established without additional infiltration of progenitors from the periphery were chosen as positive control. At the time of analysis on E21.5 microglia uniformly labeled with TOMATO, as expected. However, no TOMATO signal was detected in liver-resident MDP-like cells, monocytes and splenic monocytes. These data are included for the reviewer in Figure 1B below. Accordingly, we failed to detect TOMATO signal in splenic cDCs, although some TOMATO signal was observed in splenic macrophages. Since splenic macrophages lack *Cx3cr1* (Yona et al., Immunity, 2013), TOMATO signal must arise from expression of *Cx3cr1* in their progenitors. Thus, intra-uterine labelling of *Cx3cr1* expressing progenitors by tamoxifen was technically successful, however, in the absence of monocyte labelling, which serve as positive control for MDP labelling, it is not possible to conclude that cDCs arise independently of fetal liver monocytes. Finally, *Cxcr4^{creERT2}* mice have been shown to label consecutive stages of definitive hematopoiesis in embryonic and adult mice, while not labeling YS macrophage progenitors (Werner et al., Nat. Neurosci., 2020). We therefore used *Cxcr4^{creER}Rosa^{mtmG}* mice and pulsed with tamoxifen at E12.5 reasoning that fetal liver HSCs would be labeled, allowing us to assess their contribution to splenic DCs. We were able to perform one preliminary experiment to test this hypothesis and the data of six

embryos are provided for the reviewer in Figure 1 C below. At E18.5, microglia, which served as negative control, exhibited no GFP labeling, as expected (Werner et al., Nat. Neurosci., 2020). In contrast, we observed labelling of monocytes and macrophages in E18.5 spleen, consistent with their descent from fetal liver HSCs (Figure 1 C below). Importantly, cDCs labelled with GFP to a similar extent, supporting the notion that fetal liver-resident progenitors contribute to cDC2 in early life. We are nonetheless reluctant to include the data as is in the revised manuscript as we believe this would require confirmation in at least one additional independent litter of mice. While such data could be generated to be included in the present manuscript, it is most likely that a fetal liver-resident progenitor contributes to cDC2 in early life for the reasons already outlined above. This is now discussed on page 24 of the revised manuscript.

Figure 1. (A) Pregnant C57BL/6J mice were injected *i.p.* with 100 μ g/g of EdU on E12.5. Spleens were isolated from the mothers as well as from newborn pups at PND1. CD24⁺ and CD11b⁺ DC subsets were identified as indicated and the incorporation of EdU was analyzed by flow cytometry. Representative plots showing the detection of EdU signal in each splenic DC subset from the mother (upper row) and newborn mice (lower row). (B) Pregnant *Cx3cr1^{creER}Rosa^{Tom}* dams were injected with tamoxifen on E12.5. Spleen, liver and brain from offspring mice were analyzed by flow cytometry on E21.5. The percentage of TOMATO⁺ cells in the indicated populations is plotted. (C) Male *Cxcr4^{creER}* mice were mated with female *Rosa^{mTmG}* mice and pregnant dams were injected with tamoxifen on E12.5. Spleen and brain from offspring mice were analyzed by flow cytometry on E18.5. The percentage of GFP⁺ cells in the indicated populations is plotted. Each dot represents one mouse, horizontal bars represent mean, error bars represent SD.

3. Does the RAG1-labelled pool of precursors in the neonatal liver or bone marrow contain MDPs or CDPs?

In addition to our analysis of bone marrow progenitors in *Rag1^{Cre}Rosa^{YFP}* mice on PND7-9 and in adulthood, we have now profiled YFP expression in CLP, MDP, CDP and TER119⁺ cells in bone marrow on PND2-3. These data are provided as revised Suppl. Fig. 3C. Similar to observations on PND7-9 and adulthood, CLP strongly labelled with YFP, whereas labelling of CDP and MDP did not exceed labeling of TER119⁺ cells, which serve as negative control. Thus, MDPs and CDPs do not label in *Rag1-Cre* mice at any age examined, whereas CLPs are similarly labelled across ages. These data further support the suitability of *Rag1-Cre* mice to fate map cells of lymphoid, but not myeloid origin in early life.

4. Would a fetal DC progenitor exclusively generate TOM-cDC2 in an adult mouse?

The reviewer raises an interesting question. Since we have not formally defined ‘the’ fetal DC progenitor, we could address this experimentally by transferring unfractionated fetal liver progenitors into adult mice. However, if fetal liver progenitors generated TOM⁺ DC2 in adult mice, we would not be able to pinpoint, if this were the same progenitor population that yields TOM⁺ DC2 in neonatal mice *in vivo*. For this reason, while such data could be generated to be included in the present manuscript, the results from such line of experimentation would not alter the main conclusions of the manuscript.

5. On several occasions the authors refer to environmental signal as regulator of cDC2 function. What is the impact of the microbiome as environmental trigger on the phenotype of cDC2? Would cDC2s isolated from adult germ-free mice share or retain functional features of a neonatal cDC2?

We thank the reviewer for this comment. We have isolated cDC2 from adult germ-free (GF) and SPF mice and cultured them with OT-II cells under Th0, Th17 and Treg conditions to assess if cDC2 from germ-free mice retain functional features of neonatal cDC2 (i.e. increased Treg or Th17 differentiation capacity). These data have been included as new Suppl. Fig. 6I. In these experiments cDC2 from germ-free mice induced similar Th17 and Treg differentiation, when compared to cDC2 from SPF mice. Additionally, we observed similar induction of IFN- γ production under non-polarizing conditions in T cells cultured with cDC2 from GF and SPF mice. Thus, cDC2 from GF mice do not recapitulate the phenotype of neonatal cDC2.

GF mice have a complete absence of commensals, but in an SPF environment cDC2 are expected to encounter a variety of microbial signals within days of birth. Therefore, we believe that these observations do not exclude the possibility that microbially derived signals, or signals induced in response to commensals, play a role in imprinting cDC2 function with age. A recent report indicates that a transient wave of TNF- α induced by microbial interaction within the first days of life induces the maturation of neonatal cDC1 (Köhler et al, Gut, 2020). Similarly, a transient spike in TNF signaling induced by microbial signals during weaning has been reported to have a lasting influence on immune development (Al Nabhani et al., Immunity, 2019). Thus, age dependent differences in cDC2 function could be related quantitative/qualitative differences in the composition of the microbiome at different stages of life or related to lasting alterations on progenitor populations similar to the concept of trained immunity mentioned by the reviewer below. This is also discussed on page 27 of the revised manuscript. Investigations regarding these possibilities are currently ongoing in the lab. However, in depth analyses of the impact of microbial signals on the age-dependent functions of cDC2 will require significant effort. We therefore hope that the reviewer will agree that such data could be generated and used as the basis for a follow-up manuscript.

6. Considering the impact of microbial colonization on a plethora of immunological processes, would cDC2 isolated from adult type I interferon-deficient mice recapitulate the

phenotype of neonatal cDC2?

We thank the reviewer for this comment. In new Figures 5C and D, as well as new Suppl. Fig. 6C, and in response to reviewer 2, we now provide evidence that age-dependent features of cDC2 are regulated by type I IFN. CD38 and PD-L1 are genes regulated by type I and type II interferons that we identified to be differentially regulated with age in RNA sequencing. We now show that CD38 and PD-L1 protein expression is higher in cDC2 from adult compared to 2 week old mice (new Fig. 5C and new Suppl. Fig. 6C). In accordance with these proteins being regulated by type I interferons, we found reduced CD38 and PD-L1 protein in mice deficient for IFN- α receptor (*Ifnar*^{-/-}) compared to control mice (new Fig. 5D). Thus, cDC2 from adult *Ifnar*^{-/-} mice recapitulate the phenotype of neonatal cDC2 in terms of PD-L1 and CD38 expression, supporting the notion that age-dependent differences in type I interferon signaling impact cDC2 phenotype. Notably, cDC2 from germ-free mice do not show reduced expression of CD38 and PD-L1 compared to cDC2 from SPF mice (new Suppl. Fig. 6J), which is in line the fact that cDC2 from GF mice do not recapitulate functional features of neonatal cDC2 (see point 5 above). Taken together these data support that age-dependent differences in cDC2 phenotype are regulated, at least in part, by type I IFN. However, such age-dependent differences in IFN signaling in cDC2 appear not merely due to lower microbial load in early life. This is also discussed on pages 20 and 27 of the revised manuscript.

7.The authors report distinct expression levels of TLR4 and TLR5 on cDC2 isolated at different time points of development. Could treatment of neonatal mice with LPS or Flagellin revert the “tolerogenic” phenotype of cDC2 or influence their respond to other TLR ligands in light of trained immunity? Do these signal impact the bone marrow CDP pool?

Whether trained immunity exists for cDCs and especially for cDCs in newborn mice is an interesting question that we are currently investigating. Preliminary data from the lab indicates that cDC2 from antibiotic treated mothers show decreased cytokine production in response to TLR stimulation. These data would suggest that cDC function can be enhanced by microbially derived signals, either directly or via the action of cytokines. As such, it is likely that trained immunity takes place for cDCs in early life but investigating this point requires careful and in-depth analyses. We find higher TLR5 expression in cDC2 from adult compared to young mice, whereas TLR4 expression is higher in cDC2 from young mice. Treatment of neonatal mice with flagellin and LPS, as suggested by the reviewer, could act directly on cDC2 and modify their function, which we would need to investigate. Alternatively, it could induce a strong cytokine response in other cells that consequently could modify cDC2 function or impact DC progenitors. Distinguishing these possibilities will require in depth analyses. We can perform the suggested experiments but we hope that the reviewer agrees that an investigation into trained immunity of neonatal cDC2 could be used as the basis for future studies.

8.Even though TLR9 expression is reported to be similar in cDC2 from adult and young mice, cytokine responses are differing. These findings suggest functional differences between ontogenetically distinct DCs. Do TOM- cDC2 display phagocytic differences?

As suggested, we have assessed the phagocytic ability of TOM⁻ DC2 and these data are provided as new Suppl. Fig. 6H. TOM⁺ cDC2 and TOM⁻ DC2 from 2-week-old and TOM⁺ cDC2 from adult *Clec9a*^{cre/+}*Rosa*^{Tom} mice were cultured with fluorescently labeled latex beads *in vitro* for 2 hours in the presence or absence of cytochalasin D. We found that all populations tested were able to take up fluorescently labelled beads by phagocytosis, as bead uptake was strongly inhibited by cytochalasin D (new Suppl. Fig. 6H). Importantly, we observed no differences in the efficiency of bead uptake between TOM⁺ cDC2 and TOM⁻ DC2 from 2-week-old and TOM⁺ cDC2 from adult *Clec9a*^{cre/+}*Rosa*^{Tom} mice, indicating TOM⁺ cDC2 and TOM⁻ DC2 do not differ in their phagocytic ability. We agree with the reviewer that it is possible that functional differences exist between TOM⁺ cDC2 and TOM⁻ DC2 in situations that we have not explored in this manuscript. We now discuss this possibility of page 24 of the revised manuscript.

9. While we acknowledge that $RORc^{-/-}$ mice have several developmental defects, do TOM-cDC2 require $RORc$ for their development?

This is an interesting question. Investigating the specific dependence of TOM-DC2 on $RORc$ requires sanitation of $RORc^{-/-}$ mice into our facility (no mouse import was possible due to the Corona pandemic until recently) and at least two generations of breeding $RORc^{-/-}$ mice to $Clec9a^{Cre}Rosa^{Tom}$ mice. Instead, we have tried to address the reviewer's comment by investigating cDC development in neonatal $RORc^{-/-}$ mice to potentially correlate a partial reduction of cDC2 to a developmental dependence of TOM-DC2 on $RORc$. Unfortunately, two independent litters only generated 2 $RORc^{-/-}$ mice but their analyses indicate that the frequency of cDC2 is similar between $RORc^{-/-}$ and $RORc^{+/-}$ littermate controls (see Figure 2 below), suggesting that TOM-DC2 develop normally in the absence of $RORc$. We can repeat the experiment with more mice, however, as the reviewer acknowledges $RORc^{-/-}$ mice have several developmental defects and while correlating $RORc$ dependency of cDC2 in young mice to TOM-DC2 would be interesting, the outcome of such investigation would not alter the main conclusions of our paper.

Figure 2. Spleens from PND3-4 $Rorc^{+/-}$ and $Rorc^{-/-}$ littermate mice were analyzed by flow cytometry. After gating on single live autofluorescence negative cells, cDC2 were identified as $CD11c^{+}MHCII^{+}CD11b^{+}$ cells and their frequency in total live leukocytes is shown for each genotype. Each dot represents one mouse, horizontal bars represent mean, error bars represent SD. Data are combined from 2 independent experiments.

Minor:

Are the axis on the first dot plot in figure 1H are mislabeled?

We thank the reviewer for pointing this out and apologize for the oversight. The labeling has been corrected.

Page 8, first paragraph of section: Reference to Fig 3C should instead be Fig 2C

We thank the reviewer for pointing this out, the reference to the figure has been corrected.

Fig 4E: Since TOM- cDCs constitute a very small fraction of the adult splenic cDC2 pool (Fig 1F), it may be informative for the reader to include a representative gating strategy of the adult sample for this figure as a supplementary figure.

We thank the reviewer for this suggestion and have added a representative gating in revised Fig. 1F.

Reviewer #2 (Immune regulation, immune-stromal interaction) (Remarks to the Author):

Papaioannou et al. Age-dependent environmental signals rather than layered ontogeny imprint the function of type 2 conventional dendritic cells.

In this paper, the authors use cre-mediated fate mapping to track the progeny of $Clec9a$ -expressing dendritic cell precursors. They found that in neonatal mice, cDC2 arise from $Clec9a^{+}$ and $Clec9a^{-}$ precursors, as opposed to adult mice in which almost all splenic cDC2 arise from $Clec9^{+}$ precursors. Transcriptome analysis revealed that both neonatal cDC2 subsets greatly differ from the adult cDC2s but also that the minor variation among neonatal

cDC2 subsets could be caused by proliferating cDC2. ESAM⁺ and ESAM⁻ DCs are found in both tomato⁺ and tomato⁻ subsets, whereas RORc-expressing DCs are found only in the tomato⁻ subset in neonates. Ex-vivo co-culture of cDC2 with OT-II CD4 T cells and under polarizing conditions showed few functional differences among cDC2 subsets. Using anti-DCIR2 antibody to deliver OVA to cDC2, the authors showed a slightly higher capacity of neonatal cDC2s to induce IFN- γ production in CD4 T cells ex-vivo.

Overall, this work aimed to define the factors that drive the phenotypical and functional differences previously reported between adult and neonatal cDC2 population. Although the authors find that tomato⁺ DCs increase with time, we have no idea about the separate precursors to the tomato⁻ cells. Moreover, tomato expression is not the main distinction of function, whereas time after birth is more related to functional differences, perhaps related to functional differences between DC2 subsets that are not explored here. Thus, the authors assert that the reason behind age-dependent functional differences of cDCs is due to differences in cytokine environments that imprint cDC2 with a distinct gene expression profile, which is likely true, but they did not provide any evidence to support their statement. What cytokines? Where? In the end, we know that the frequencies of various DC populations change over time, but we don't know how or why or whether these subsets are functionally distinct at the different times.

We thank the reviewer for this concise summary. We would like to clarify, however, that a detailed characterization of splenic cDC2 with age has not previously been performed and the demonstration that cDC2 in early life are not an “immature version” of adult cDC2 presents a main novelty of our studies. Additionally, we describe for the first time that cDC2 develop in waves. That cell origin may not be the main determinant for cDC2 function may be perceived as a contradiction but specifically the question whether cell origin (nature) or environmental signals (nurture) influence cell identity remains a fundamental debate in dendritic cell biology. The developmental heterogeneity of early life cDC2 was revealed using *Clec9a-Cre* mice, which, despite possible shortcomings that we discuss and experimentally address in the revised manuscript, provide the state-of-the-art model to fate map conventional dendritic cells within the realm of the highly competitive field of mononuclear phagocyte biology. We provide new data in the revised manuscript to highlight the faithfulness of the *Clec9a-Cre* system in early life. Importantly, our conclusions towards the developmental heterogeneity of cDC2 in early life are not solely based on *Clec9a^{cre}* fate mapping but substantiated by independent fate mapping models. In the revised manuscript we have further validated the lymphoid contribution to the neonatal but not adult cDC2 compartment using *Il7r-Cre* mice. The specific concerns of the reviewer have been addressed as outlined in the point by point response below.

1. A major point of the manuscript is that many of the neonatal cDC2 cells are derived from different (CLEC9⁻) precursors than the adult cDC2 cells, which are mostly derived from CLEC9⁺ precursors. Despite this difference, the resulting cDC2 cells are phenotypically and transcriptionally identical. An alternative to this surprising result is that neonatal and adult cDC 2 cells are derived from exactly the same precursors and that the only difference is the efficiency of recombination – perhaps because the cDC2 cells in neonates transition very rapidly through the precursor stage. Is tomato expression in the pre-cDC cells (gated as in Fig 1 G-H) the same in neonates and adults?

As suggested and in response to reviewer 3, we have profiled TOMATO expression in progenitors for DCs and other lymphoid and myeloid lineages in neonatal mice. The data are provided as new Suppl. Fig. 2E. and further highlight the faithfulness of *Clec9a^{Cre}* mice to fate map cDCs in early life. TOMATO was detected in CDP and pre-cDC-like cells, but not MDPs or CLPs from one and two-week old, as well as adult, *Clec9a^{cre/cre}Rosa^{TOM}* mice (new Suppl. Fig. 2E). In this context we would like to point out that we employed a phenotypic definition based on surface markers that identify these cells in

adulthood and thus refer to CDP-like and pre-cDC-like cells in neonates. Additionally, the only definitive marker to distinguish cDC restricted progenitors from MDPs and other progenitors, DNGR-1/CLEC9A, cannot be detected in *Clec9a^{cre/cre}* mice, which are a knock out for CLEC9A (Schraml et al., Cell, 2013). Thus, the defining gates between progenitors, especially MDP and CDP, may not be absolute, which could affect TOMATO labelling. Importantly, pre-cDC labelling reached adult levels by two weeks of age, whereas on day 7 after birth pre-cDC labelling had not quite reached adult levels (% pre-cDC labelling: 81.15±2.77% PND7, 94.67±1.72% 2weeks, 93.61±2.93% adult). At all ages examined pre-cDC labelling in spleen and bone marrow was similar, indicating that Cre-mediated labelling of pre-cDCs completes in bone marrow. In line with our previous observation (Schraml et al., Cell, 2013), CDP labelling was lower than that of pre-cDCs in adult mice and in early life. These data confirm that *Clec9a*-driven Cre was active in DC precursors in early life.

If cDC2 in neonates escaped labelling because of a rapid transition through the precursor stage, we would expect higher labelling in the differentiated cDC2 than in their progenitors. However, in one and two week old mice, TOMATO labelling in pre-cDCs exceeded that of splenic cDC2, which is in contrast to adult mice, in which pre-cDCs and cDC2 labelled to the same extent. Coupled to the observation that TOM⁻ DC2 do not acquire TOMATO after a period in culture (Suppl. Fig. 2G), these data further support, that the relative lack of cDC2 labelling in neonatal mice is not an ‘artifact’ due to “escape“ from labelling but due to the existence of an alternative route to cDC development. As pointed out by the reviewer escape from recombination is an inherent caveat to lineage tracing. For this reason, we have performed fate mapping of DCs in alternative models to validate the developmental heterogeneity of neonatal cDC2. Fate mapping in *Rag1-Cre* and new data from *Il7r-Cre* mice, provided in response to point 2 below and response to reviewer 3, support a lymphoid origin. As such, the finding that TOM⁺ cDC2 and TOM⁻ DC2 are transcriptionally identical is not surprising but in line with previous reports showing that cDC-like cells arising from lymphoid progenitors are virtually indistinguishable from *bona fide* myeloid derived cDCs (Helft et al., Cell Rep, 2017 and Salvermoser et al., Front Immunol, 2018). Importantly, we demonstrate for the first time a physiological situation in steady state when lymphoid progenitors contribute to DC-poiesis.

2. Although the authors claim that the neonatal cDC2 cells come from different precursors, they seem to be uniformly dependent on FLT3 and Myb, but independent of CSF1 (Fig 2) and have similar phenotype and placement (fig 3), again suggesting that tomato labeling is not really a distinguishing feature and may be an artifact.

The ability to distinguish ontogenetically distinct cell types, when all other parameters are equal is an inherent strength of fate mapping. In response to the concerns raised in point 1 above, we now provide additional data to highlight the faithfulness of *Clec9a-Cre* mediated fate mapping in early life. To exclude the possibility that lack of TOMATO labelling in cDC2 of *Clec9a^{Cre}Rosa^{Tom}* mice in early life is an artifact, we have confirmed a developmental heterogeneity in early life cDC2 independent fate mapping models. Using *Rag1^{Cre}* mice we have found a lymphoid contribution to cDC2 in early life but not adulthood. In the revised manuscript we have further substantiated these observations using interleukin 7 receptor (Il7r) – Cre mice as an alternative model to fate map cells of lymphoid origin (Schlenner et al., Immunity, 2010). *Il7r^{Cre}* mice have previously been used to demonstrate that lymphoid progenitors do not contribute to cDCs in adult steady state mice (Schlenner et al., 2010). We have crossed IL-7R^{Cre} mice to Rosa^{lox-stop-lox}-red fluorescent protein (RFP) reporter mice and profiled RFP expression in splenic populations from 2-3-day and 7-9-day old mice. These data have been added as new Fig. 2E. As expected, B cells and pDCs showed near complete labelling with RFP at both ages examined (>94% RFP positive), whereas neutrophils, which served as negative control were poorly labelled in 2-3 day-old (18.5±3.38%) and 7-9-day old mice (4.54±0.75%). Importantly, cDC2 labelling exceeded that of neutrophils at both time points. In 2-3-day old mice cDC2 labelled at 70.6±5.11%, whereas labelling decreased to 53±2.05% in 7-9-day old mice. Therefore, analogous to our findings in *Rag1-Cre* mice, fate mapping in *IL7R-Cre* mice demonstrates a lymphoid contribution to cDC2 in early life that disappears with age. Thus, the lack of Tomato labelling of cDC2 of neonatal *Clec9a^{Cre}Rosa^{Tom}* mice is not an artifact but a true indicator of developmental heterogeneity as confirmed in two independent fate mapping models.

3. The authors sorted DCs based on tomato expression and evaluated T cell responses. Most differences were related to age rather than the expression of tomato. Of course we know that the composition of cDC2 cells changes over time based on ESAM expression and that ESAM does not correlate with tomato expression, but that ESAM⁺ and ESAM⁻ DCs have different transcriptional programs and stimulate different types of T cell responses. Do ESAM⁺ and ESAM⁻ cDC2 cells have different functions with age?

We thank the reviewer for this comment. The number of cDC2 that can be isolated from spleen in early life is limiting, leading us to address functional differences of cDC2 with age using bulk cDC2. As discussed on Page 19 of the revised manuscript, we chose a time point when ESAM^{high} to ESAM^{low} subset distribution had reached adult levels and ESAM^{high} cells dominate the cDC2 compartment to account for putative functional differences between ESAM^{high} to ESAM^{low} cDC2. We now compare ESAM^{high} to ESAM^{low} cDC2 across age in new Fig. 5H. We sorted ESAM^{high} and ESAM^{low} TOM⁺ cDC2 from two-week-old and adult mice and assessed their ability to stimulate Th17 and Treg differentiation. Due to the limiting number of DCs at two weeks of age, we were not able to sort sufficient TOM⁻ ESAM^{low} DC2 to allow for comparison of TOM⁻ ESAM^{high} and ESAM^{low} cells. When comparing ESAM^{high} and ESAM^{low} cDC2 across age, both populations from young mice induced higher Th17 differentiation than their counterparts from adult mice. Similarly, ESAM^{high} cDC2 from young mice induced higher Treg differentiation than their adult counterparts, whereas ESAM^{low} cDC2 from young and adult mice induced similar Treg differentiation. Thus, both ESAM^{high} and ESAM^{low} cDC2 exhibit functional differences with age, although there appears to be a level of subset specific functional regulation. This is also discussed on pages 19, 20 and 26 of the revised manuscript.

4. The other difference between neonatal and adult cDC2 cells is the presence of the RORc-expressing population 11 that is highest in neonatal tomato- cells and essentially not present in adult cells. These are also functionally distinct, and differ in frequency in the tomato⁺, tomato⁻ and the adult cDC2 populations. The frequency of these cells changes over time – does their function change? Without accounting these differences in composition, I cannot interpret the T cell responses.

As pointed out by the reviewer, ROR γ ⁺ DC2 are exclusively found in the TOM⁻ fraction of cDC2 and exclusively present in early life. As such, they do not influence any functional comparison of Tom⁺ cDC2 across age. At two weeks of age these cells constitute less than 2% of the TOM⁻ fraction of cDC2 (Fig. 4E), rendering it highly unlikely that they alter functional readouts. The reviewer is correct, however, in pointing out that ROR γ ⁺ TOM⁻ DC2 could influence functional read outs as they are transcriptionally unique. We now provide new data in response to reviewer 3 in new Suppl. Fig. 4C that demonstrate that ROR γ ⁺ TOM⁻ DC2 are predominantly ESAM^{high} by flow cytometry, consistent with their transcriptional similarity to ESAM^{hi} cDC2 (Fig. 4F and Suppl. Fig. 5F). As such, ROR γ ⁺ DC2 could influence functional read outs of the ESAM^{hi} TOM⁻ DC2 population. To address this possibility we crossed *Rorc-eGFP* mice to *Clec9a^{cre/cre}Rosa^{TOM}* mice. We then sorted ESAM^{high} TOM⁺ cDC2 and ESAM^{high} TOM⁻ cDC2 from two-week-old *Rorc^{eGFP}Clec9a^{cre/+}Rosa^{TOM}* mice while excluding GFP⁺ cells and compared their ability to promote Th17 and Treg differentiation of OT-II cells. In this set up, ESAM^{high} TOM⁻ cDC2 and ESAM^{high} TOM⁺ cDC2 had similar ability to induce Th17 and Treg differentiation (new Suppl. Fig. 6G). Thus, the presence of ROR γ ⁺ cells does not significantly alter the ability TOM⁻ DC2 to induce T cell differentiation.

5. Figure 1H - X axis should be CD43?

We thank the reviewer for pointing this out. It has been corrected.

6. Fig 4F – it would help the reader a great deal if this panel (or a copy of this panel) had

outlines surrounding the clusters with a callout as to their identity. For example, clusters 1, 2, 4, 9 and 13 can be outlined together and labeled as cDC1 cells, whereas 0, 3, 7 and perhaps 11 can be outlined together and called out as DC2 cells.

We thank the reviewer for this suggestion. The figure has been revised accordingly.

Reviewer #3 (DC development, single cell analysis) (Remarks to the Author):

Summary:

The study of Papaioannou et al characterizes the development and lineage affiliation of conventional dendritic cells (cDCs) throughout neonatal and adult life. Using the *Clec9a* lineage tracer mouse, the authors show that cDC2 are initially generated from *Clec9a*-progenitors which are progressively replaced by *Clec9a*+ CDP derived cDC2s. From RNA Sequencing analysis they show that age appears being the major driver for differential gene expression as well as function within the cDC2 subsets.

Originality and significance:

The authors use the myeloid specific *Clec9a* Cre lineage tracer mouse model to assess the developmental origin and the possible functional differences of splenic cDC2 from early postnatal age to adulthood. It had been previously observed that fetal DCs are less capable of priming and show a Th2 bias. As no major difference could be phenotypically detected the authors perform bulk as well as single cell RNA sequencing. The major finding resides in the observation that cDC2 derived from young mice despite showing a reduced inflammatory profile are capable of inducing a stronger Th response which is polarized towards Treg and Th17.

The observation and the dataset are novel, however more experiments would be required to validate the findings.

Major Points:

1. The authors used the *Clec9a*-Cre x *Rosa26LSL*-TOM lineage tracer model to assess the developmental history of cDC2 in neonatal as well as adult mice. It would be essential to include a detailed analysis of the labeling efficiency across myeloid and lymphoid progenitors (MDPs, CDP, pre-DCs, CLP) across the first days/weeks of age. In particular three studies have recently shown the contribution of lymphoid progenitors to the cDCs and pDCs pool (Ref 74- Ref 112, please include Rodrigues et al.) It would be important to include in the study pDCs as those were suggested to be mostly (Ref 74/ Rodrigues et al.) or exclusively (ref. 112) lymphoid derived and would therefore validate the *Clec9a*+ CDP origin of adult cDC2 versus CLP or *Clec9a*- CDP derived Young cDC2.

In revised Suppl. Fig. 2E we have now provided the labelling efficiency of MDPs, CDPs, pre-cDCs and CLPs in bone marrow of one and two week old *Clec9a^{cre/cre}Rosa^{TOM}* mice. In this context we would like to point out that we refer to these cells as CDP and pre-cDC-like cells because we defined these cells based on the same surface marker phenotype as in adulthood. Additionally, use of DNGR-1/CLEC9A, which is the only definitive marker to distinguish cDC restricted progenitors, especially from MDP, is not possible in *Clec9a^{cre/cre}* mice, which are a knock out for CLEC9A (Schraml et al, Cell, 2013). TOMATO signal was detected in CDPs and pre-cDCs, but not MDPs of one week old and two week old *Clec9a^{cre/cre}Rosa^{TOM}* mice (new Suppl. Fig. 2E). No TOMATO labelling was detected in CLPs from one week old and adult *Clec9a^{cre/cre}Rosa^{TOM}* mice and TOMATO was also absent from the lin⁻CD117^{int}CD115⁻CD135⁺ fraction of bone marrow from two week old mice, which includes CLPs (Onai et al., Nat. Immunol. 2007). These data confirm that *Clec9a*-driven Cre was active in DC precursors

but not MDPs or CLPs in early life. Additionally our analysis revealed that at all ages examined pre-cDC labelling in spleen and bone marrow was similar, indicating that cre-mediated labelling of pre-cDCs completes in bone marrow. If cDC2 in neonates escaped labelling because of a rapid transition through the precursor stage, we would expect higher labelling in the differentiated cDC2 than in the progenitor. Importantly, labelling of pre-cDC exceeded that of splenic cDC2 in early life, whereas in adult mice pre-cDCs and cDC2 labelled to the same extent. As such, these data further support, that the relative lack of cDC2 labelling in neonatal mice is not attributed to progenitors that escape labelling and further support the faithfulness of *Clec9a*Cre to fate map cDCs in early life.

We thank the reviewer for pointing out that we had omitted to cite Rodrigues et al. This reference has now been added. We agree with the reviewer that pDCs serve as a good reference population for lymphoid origin. However, they cannot be used as reference in *Clec9a^{Cre}Rosa^{Tom}* mice because pDCs express DNGR-1 in their differentiated form and therefore label with TOMATO in these mice. We have however, included pDCs as reference population for lymphoid origin in *Rag1^{Cre}* and *Il7r^{Cre}* mice (revised Fig. 2D and new Fig. 2E). pDCs show similar labelling to B cells in *Il7r^{Cre}* mice at two-days and one week after birth (Schlenner et al., Immunity, 2010). Labelling of pDCs in *Rag1Cre* mice is lower than that of B cells but labelling of pDCs is relatively constant across different ages in both models. Importantly, labelling of pDCs exceeds that of cDC2 in both models and at all ages examined. These data therefore validate that a lymphoid contribution to the neonatal but not adult cDC2 compartment.

2. The heterogeneity across the used models Tomato -Cre; Cre homozygous and YFP-Cre, should be discussed in more detail as the different ontogeny based on *Clec9a* is a major point of the study.

We thank the reviewer for pointing out that we were not sufficiently clear on this point. As discussed on page 7 of the revised manuscript we have previously shown that higher levels of Cre expression (in homozygous *Clec9a^{Cre}* mice) leads to markedly increased labeling of DC precursors and differentiated cDCs because a fraction of precursors “escapes” labeling in a stochastic fashion (Schraml et al, Cell 2013). Additionally, the choice of fate reporter can influence labelling efficiency due to differences in the distance between the loxP sites. We therefore wanted to ensure that reduced labeling of neonatal cDC2 was not an artifact of the specific fate reporter used or due to inefficient recombination in progenitors escaping labelling in mice heterozygous for Cre. Importantly, the conclusion that neonatal exhibit ontogenetic diversity is not only based on *Clec9a^{Cre}* -mediated fate mapping, but was further confirmed using *Rag1^{Cre}* and *Il7r^{Cre}* mice (new Fig. 2 E) as two independent fate mapping models. For functional assays we used mice heterozygous for Cre whenever possible, because *Clec9a^{Cre}* mice are a knock-out for CLEC9A. Although CLEC9A has no known function in DC development and is not expressed in cDC2, we nonetheless wanted to exclude any artifacts in these assays due to lack of DNGR-1. We have clarified our reasons for using the different mouse models throughout the revised manuscript.

3. Rag expression from Rag-Cre x Rosa26LSL-YFP is here used to trace lymphoid derived cDC2, how is the the labelling distributed across ESAM+ and ESAM- cDC2 subsets?

YFP labelling of ESAM^{high} and ESAM^{low} cDC2 subsets in one-week old *Rag1^{Cre}Rosa^{YFP}* mice is similar. These data has been included in the manuscript as new Suppl. Fig. 3D. Together with fate mapping in *Clec9a^{Cre}* mice (Suppl. Fig. 1E) these data support that ESAM^{high} and ESAM^{low} cDC2 exhibit a similar developmental heterogeneity in early life.

4. Proliferation related genes appear to be the mostly differentially expressed genes that characterize young mice. What is the survival in culture of Adult versus young DCs. Could the better priming capacity of Young DCs be explained by higher proliferation and survival in culture?

At the end of a 3,5 day culture, we find too few DCs alive in all conditions to answer this question. We could set up short term DC cultures to assess if cDC2 from adult and young mice exhibit differences in survival/proliferation, however, we do not believe that such differences explain the increased Th17 and Treg differentiation capacity of cDC2 from young mice. If increased survival or proliferation would play a role, we would expect differences in OT-II proliferation and differentiation in all conditions examined. Instead, we find that young vs. adult cDC2 have similar ability to induce T cell proliferation under Th0, Th1, Th17 and Treg conditions, as assessed by CTV dilution. The fact that cDC2 from young mice specifically induce higher Th17 and Treg differentiation, but not Th1 differentiation, compared to cDC2 from adult mice indicates qualitative, rather than quantitative differences in the ability to polarize T cells. This is now discussed on page 19 of the revised manuscript.

5. The authors identified a Rorgt+ cDC2 subset within the Tom- fraction. A similar subset was recently described by the group of Rudensky (ref. 89). It would be relevant to include a detailed phenotypic characterization of the subset (CD11b, Clec4a, Sirpa, CD26, ESAM, CD117, CD80, MHC-II, CD24).

As suggested, we now provide a detailed phenotypic analyses of these cells in new Suppl. Fig. 4C,D. ROR γ ⁺ TOM⁻ DC2 and TOM⁺ cDC2 express similar levels of MHCII, CD11b, CD26, CD24, as well as CD80 and CD86, whereas expression of CD172a is slightly lower on ROR γ ⁺ TOM⁻ DC2 than on TOM⁺ cDC2. Notably, ROR γ ⁺ DC2 expressed higher levels of CD117 and CLEC4A4 than cDC2 and uniformly expressed high levels of ESAM. In line with transcriptional analysis, ROR γ ⁺ TOM⁻ DC2 therefore more closely resemble ESAM^{high} cDC2 (Suppl. Fig. 5F). As such, ROR γ ⁺ TOM⁻ DC2 appear to be distinct from the cDC2B population recently described by Brown et al., because cDC2B lack ROR γ t protein and transcriptionally resemble ESAM^{low} cells. Based on signature gene expression, ROR γ ⁺TOM⁻DC2 additionally more closely resembled cDC2A than cDC2B (Suppl. Fig. 4, 5) and ROR γ ⁺TOM⁻DC2 appear absent from adult spleen. This is also discussed on pages 14 and 16 of the revised manuscript. Taken together phenotypic profiling shows that ROR γ ⁺TOM⁻DC2 express prototypical cDC2 markers and further support that these cells constitute a unique subset of cDC2.

6. It would be important to clearly mention that the Bulk RNA sequencing is performed at 8 days of age and therefore as shown young as well as adult derived cDC2 subsets differ in their composition especially in the % of ESAM expressing cells, please refer to the related supplementary data. Similarly, in the T cell stimulation experiments cDCs from a 2.5 weeks of age mice are used, again it is important to point out to the exact composition of the cDC2 compartment (Figure 1E, 4 and 5). This could also be the reason for the PCA segregation of adult DCs, which would include a subset absent in young mice.

We have now clarified the reasons for choosing different ages of mice and pointed out the exact composition of the cDC2 compartment throughout the manuscript. We acknowledge that some of the age dependent differences observed in RNAseq could be due to differences in the composition of cDC2 with age. In response to reviewer 2, we have added a functional comparison of ESAM^{high} and ESAM^{low} cDC2 across age, which is included in the revised manuscript as new Fig. 5H. We sorted ESAM^{high} and ESAM^{low} TOM⁺ cDC2 from two-week-old and adult mice and assessed their ability to stimulate Th17 and Treg differentiation (new Fig. 5H). We found that both ESAM^{high} and ESAM^{low} cDC2 from young mice induced higher Th17 differentiation than their counterparts from adult mice. Similarly, ESAM^{high} cDC2 from young mice induced higher Treg differentiation than their adult counterparts, whereas ESAM^{low} cDC2 from young and adult mice induced similar Treg differentiation. These data establish that ESAM^{high} and ESAM^{low} cDC2 exhibit similar age-dependent functional differences, although there appears to be an additional level of subset specific functional regulation.

7. The authors elegantly overcome the problem of target a specific subset by delivering antigen and subsequently sorting individual subsets. Could the delivery of Antigen loaded DCs from Young versus Adult mice reproduce the results obtained?

The question whether *in vitro* antigen loaded DCs would induce T cell responses upon adoptive transfer is interesting and we would expect so. In the current manuscript wanted to investigate whether cDC2 from early and adult life have the capacity to induce T cell differentiation upon antigen delivery *in vivo*, which demonstrated using the antigen targeting approach. As such, adoptive transfer of antigen-load DCs would not influence the main conclusions of the manuscript

However, we believe that a major problem in the T cell experiments performed *in vitro* by the authors is the inclusion of polarizing cytokines in the assay, which could have a dominant effect. We would therefore suggest to repeat the experiments without adding polarizing cytokines. A conclusion of the data which show that increased Treg and Th17 polarization is the consequence of increased IL6 and IL10 production would be important.

The reviewer might have overlooked this experiment. In Fig. 5D, F we have assessed the ability of young and adult cDC2 to induce T cell proliferation under non-polarizing (Th0) conditions. In Th0 conditions TOM⁺ cDC2 and TOM⁻ DC2 from young mice and TOM⁺ cDC2 from adult mice induced similar proliferation and effector differentiation of OT-II cells, as assessed by CTV dilution and IFN- γ production, respectively. Under these conditions, we did not observe induction of Foxp3 positive or IL-17A producing T cells.

We would also like to clarify that although we observed increased IL-6 and IL-10 production in cDC2 from young compared to adult mice upon CpG stimulation, we do not believe these differences in cytokine production can be correlated to the increased Th17 and Treg polarization by cDC2 from young mice under skewing conditions. First, CpG was not used to stimulate DCs in these experiments and second, we would expect the addition of excess amounts of recombinant cytokines to have a dominant effect over DC-derived cytokines. Rather, we believe that age-dependent differences in T cell polarization under skewing conditions are caused by differences in expression of co-stimulatory molecules and other accessory factors that influence T cell differentiation (discussed on page 26 of the revised manuscript).

Minor Points:

There is a mistake in the gating strategy shown in Fig 1H and Suppl. Fig2D. The x axis of the second panel should be CD43 and not MHC-II.

We thank the reviewer for pointing this out. This has been corrected.

The heatmap in Figure 4G is of very poor quality and does not provide much information. Please improve the quality of this figure by adding additional markers and increasing the quality.

The figure has been amended with additional markers. We have additionally increased the quality of the picture, although we cannot exclude the possibility that quality is lost during the file conversion at Nature Communications.

REVIEWERS' COMMENTS

Reviewer #1 (Remarks to the Author):

I thank the authors for their extended additional work, especially considering the restrictions due to COVID-19. I agree with most of the authors explanations and appreciate the new data. I feel that the CXCR4-Cre fate mapping experiment should be included in the paper to support the claims. If the authors could add another set of mice, which they might have analyzed in the meantime, this would certainly add strength. I otherwise have no additional comment and congratulate the group to this beautiful work and the possible future projects arising from it.

Reviewer #2 (Remarks to the Author):

I appreciate that the authors did a lot of work using the fate mapping system using clec9-cre and other markers to define the differences in ontogeny between neonatal and adult DCs. And yet, I am not convinced that we have learned very much. Cells that are labeled with clec9-cre increase over time suggesting layered ontogeny, but in the end, those cells are essentially identical. I previously suggested that perhaps clec9 is not a faithful reporter and the authors have convinced me that it is. But perhaps the expression of clec9 is not meaningful in the early life development of cDC2 cells. In other words, development is the same, but clec9 is not a marker of those progenitors. In some ways, fate mapping in neonates is problematic as even 20% of the neutrophils are derived from rag1+IL-7R+ cells.

IN the end, the tomato+ and tomato- cells have essentially the same function and even ESAM expression does not really divide the cells into functional populations. Instead, the driver of functional differences is age. As the authors say, the age-dependent environment must be responsible for the functional differences in cDC2 cells, but we still have not idea how that effect is caused.

Reviewer #3 (Remarks to the Author):

The authors did a lot of work to improve the manuscript following all reviews comments. It is to my knowledge the first report showing age related development of DCs and in its current format a beautiful study.

We thank the reviewers for their support and encouraging comments and provide a point by point response to the specific comments below.

Reviewer comments in black

Authors response and changes to the manuscript blue.

Reviewer #1 (Remarks to the Author):

I thank the authors for their extended additional work, especially considering the restrictions due to COVID-19. I agree with most of the authors explanations and appreciate the new data. I feel that the CXCR4-Cre fate mapping experiment should be included in the paper to support the claims. If the authors could add another set of mice, which they might have analyzed in the meantime, this would certainly add strength. I otherwise have no additional comment and congratulate the group to this beautiful work and the possible future projects arising from it.

We thank the reviewer for the encouraging comments. The data from the CXCR4-CreERT2 fate mapping have been added as new Supplementary Figure 3E.

Reviewer #2 (Remarks to the Author):

I appreciate that the authors did a lot of work using the fate mapping system using clec9-cre and other markers to define the differences in ontogeny between neonatal and adult DCs. And yet, I am not convinced that we have learned very much. Cells that are labeled with clec9-cre increase over time suggesting layered ontogeny, but in the end, those cells are essentially identical. I previously suggested that perhaps clec9 is not a faithful reporter and the authors have convinced me that it is. But perhaps the expression of clec9 is not meaningful in the early life development of cDC2 cells. In other words, development is the same, but clec9 is not a marker of those progenitors. In some ways, fate mapping in neonates is problematic as even 20% of the neutrophils are derived from rag1+IL-7R+ cells.

We are happy that we have convinced the reviewer of the faithfulness of the *Clec9a*^{cre} fate mapping system. As pointed out by the reviewer, cDC2 in early life have a dual origin, arising on the one hand from *Clec9a*-negative progenitors, as well as from *Clec9a*-positive progenitors. Our data, using several independent fate mapping models, show that these *Clec9a*-negative progenitors are likely lymphoid progenitors. The fact that these developmentally distinct cDC2 in early life, which we can distinguish thus far only by fate mapping, are transcriptionally and functionally similar, is one important conclusion of our work.

Most lineage decisions are stochastic and therefore fate mapping with constitutive Cre is seldom absolute and must be considered at the population level and in the context of positive and negative control populations. Thus, while it is true that we observed some labelling in neutrophils of *Rag1*^{cre} (11±2.86% PND2-3, 6.63±2.32% PND7-9, 2.27±0.82% adults, Fig. 2D) and *Il7*^{cre} mice (18.5±3.38% PND2-3 and 4.54±0.75% PND7-9, Fig. 2E) labelling of cDC2 clearly exceeded that of these negative control populations (cDC2 label on PND2-3 at 70.6±5.11%, on PND7-9 with 53±2.05% in *Il7*^{cre} mice and on PND2-3 with YFP at 29±5.30%, on PND7-9 at 13.43±3.16% in *Rag1*^{cre}*Rosa*^{YFP} mice). We have clarified this point on page 8 of the revised manuscript.

IN the end, the tomato+ and tomato- cells have essentially the same function and even ESAM expression does not really divide the cells into functional populations. Instead, the driver of functional differences is age. As the authors say, the age-dependent environment must be responsible for the functional differences in cDC2 cells, but we still have not idea how that effect is caused.

The conclusion that layered development generates phenotypically, functionally and transcriptionally cDC2 is one important finding of our study. As discussed on page 19 of the revised manuscript, we believe that the dual origin of cDC2 in early life could serve to generate a fully functional DC compartment, at a time when bona fide DC poeisis is not fully established. However, it is equally possible that functional differences exist between TOM⁺ cDC2 and TOM⁻ DC2 in situations that we have not explored in this manuscript. We also discuss this possibility of page 19 of the revised manuscript.

Whether ESAM expression functionally divides the cDC2 population was not addressed here. We merely divided cDC2 in young and adult mice into ESAM^{high} and ESAM^{low} cells to demonstrate that functional differences between early and adult life exist for both ESAM^{high} and ESAM^{low} cDC2, allowing us to conclude that the age-dependent functional differences are not due to differences in cDC2 subset distribution in early and adult life.

We provide evidence that the environmental signals in early and adult life that imprint the cDC2 function are cytokines. This conclusion is based on transcriptional analyses that showed an enrichment of genes involved in signaling downstream of IFN- γ , TNF- α , IL-2 and IFN- α in cDC2 from adult mice (Fig. 5B). To support this conclusion, we added new data in response to reviewer 1. We show that protein expression of CD38 and PD-L1, which are genes regulated by type I and type II interferons, is higher in cDC2 from adult compared to 2 week old mice (Fig. 5C and Suppl. Fig. 6C), reflecting regulation at the RNA level (Supplementary Table 3). In accordance with these proteins being regulated by type I interferons, we found reduced CD38 and PD-L1 protein on cDC2 from mice deficient for interferon alpha receptor (*Ifnar*^{-/-}) compared to cDC2 from control mice (new Fig. 5D).

Thus, cDC2 from adult *Ifnar*^{-/-} mice recapitulate the phenotype of cDC2 from young mice in terms of PD-L1 and CD38 expression, supporting the notion that age-dependent differences in cDC2 phenotype are regulated, at least in part, by type I interferon. This is also discussed on page 13 of the revised manuscript. In this context it is interesting to note that reduced expression of CD38 and PD-L1 was not observed on cDC2 from germ-free mice compared to SPF mice (new Suppl. Fig. 6J), which is in line the fact that cDC2 from GF mice do not recapitulate functional features of neonatal cDC2. Taken together age-dependent differences in IFN signaling in cDC2 appear not merely due to lower microbial load in early life. This is also discussed on pages 16 and 22 of the revised manuscript.

Reviewer #3 (Remarks to the Author):

The authors did a lot of work to improve the manuscript following all reviews comments. It is to my knowledge the first report showing age related development of DCs and in in its current format a beautiful study.

We thank the reviewer for the encouraging comments and the suggestions to help improve the manuscript.